behaviour, cognition

ultimatum game, leverage, prosociality, chimpanzee, bonobo

**Author for correspondence:**
Alejandro Sánchez-Amaro
e-mail: alex_sanchez@eva.mpg.de

# Chimpanzees and bonobos use social leverage in an ultimatum game

Alejandro Sánchez-Amaro[1,2] and Federico Rossano[2]

[1]Department of Comparative Cultural Psychology, Max Planck Institute for Evolutionary Anthropology, Leipzig, Germany
[2]Department of Cognitive Science, University of California San Diego, San Diego, CA, USA

AS-A, 0000-0002-7464-9553

The ultimatum game (UG) is widely used to investigate our sense of fairness, a key characteristic that differentiates us from our closest living relatives, bonobos and chimpanzees. Previous studies found that, in general, great apes behave as rational maximizers in the UG. Proposers tend to choose self-maximizing offers, while responders accept most non-zero offers. These studies do not rule out the possibility that apes can behave prosocially to improve the returns for themselves and others. However, this has never been well studied. In this study, we offer chimpanzee and bonobo proposers the possibility of taking into account the leverage of responders over the offers they receive. This leverage takes the form of access to alternatives for responders. We find that proposers tend to propose fairer offers when responders have the option to access alternatives. Furthermore, we find that both species use their leverage to reject unequal offers. Our results suggest that great apes mostly act as rational maximizers in an UG, yet access to alternatives can lead them to change their strategies such as not choosing the self-maximizing offer as proposers and not accepting every offer higher than zero as responders.

## 1. Introduction

When we interact in market-based transactions governed by the law of supply and demand, we frequently have access to unilateral alternatives or assets that confer bargaining power over others. The use of alternatives that serve us to increase our advantage over a partner can be interpreted as leverage, a way to create bargaining power through the possession of resources that cannot be taken by force [1–3].

How human adults use leverage has been a topic of investigation in behavioural game theory through games such as the battle of sexes [4,5] or the ultimatum game (UG) [6–9]. For instance, in a classic UG, the proposer can freely decide how to divide a finite amount of resources between herself and a partner. The partner, referred to as the responder, can accept or reject the offer. In the latter case, if the responder rejects the proposer's offer, both of them obtain no rewards. Usually, human adults offer around 40% of the initial resources, and responders reject offers below 20%, although an increasing amount of evidence highlights significant cultural differences among human populations [10–13]. It has also been found that responders with a positive alternative outcome in a UG are more likely to reject the proposer's offer and demand more from the proposer. In such a situation, the responders have some leverage over the proposers [14].

From a comparative evolutionary perspective, using the UG can tell us whether other animal species share prosocial motivations with humans and whether they are aversive towards an inequitable outcome. That is, we can investigate whether proposers would make fair offers or would instead try to maximize their gains, and whether responders would accept unequal/unfair offers or not. By presenting our closest living relatives, chimpanzees and bonobos, with scenarios modelled after the UG, we can shed light on the nature of

human fairness. Furthermore, UGs can complement observational studies [15] to help us investigate whether non-human primates use leverage to maximize their gains.

To our knowledge, four different studies have presented great apes with different versions of the UG (e.g. UGs with two pre-established reward constellations for the proposer to choose among—also called mini UG) [16–19]. The main finding from this set of studies is that apes, in general, behave as rational maximizers when offers involve direct access to food rewards [16–18,20,21]. That is, proponents usually chose the offer that benefitted them the most, and responders accepted all favourable offers (i.e. any non-zero offer). Chimpanzees even accepted zero offers [17], thus showing no aversion towards inequitable outcome distributions. One possibility, according to Bueno-Guerra et al. [16], is that the rejection of unfair offers in a UG might only be present in societies with established norms about how to divide windfall rewards and whose members have a deep understanding of the concept of fairness—although whether non-human animals are averse towards inequitable outcomes is currently debated [22,23].

By contrast to these results, Proctor et al. [19] found that chimpanzees acted prosocially towards partners by proposing fair offers in their version of the UG (see also [24]). In their study, chimpanzees proposed tokens instead of food. Responders could then exchange those tokens for food rewards for both chimpanzees. It is thus possible that in their UG, responders' choices were more salient for proposers: since responders had to return the token to the experimenter actively, proposers may have perceived that responders had more control over the situation. Nonetheless, responders still accepted all proposers' offers—even more than in other UGs [18,20]. In general, the results from these studies seem to suggest that apes do not show social leverage understanding since responders never seem to reject positive offers, a strategy that could be interpreted as a way to influence proposers' future decisions.

One possibility is that in these studies, rejection was generally difficult for apes since it was based on inaction from the responders' side. Thus, one way to study how apes may use leverage strategically is to provide them with scenarios in which they can decide between accepting and rejecting offers through access to unilateral alternatives, as in a study by Sánchez-Amaro et al. [25]. In that study, the authors investigated whether chimpanzees would use social leverage strategically to maximize their outcomes during dyadic interactions. The authors presented chimpanzees with an apparatus representing an unequal reward distribution accessible to both individuals. To obtain the high-value reward, individuals had to wait for their partner to act before them. In the critical condition, the experimenters provided one of the two chimpanzees, the subject, with alternative access to a safe mid-value reward. The subject could then decide whether to wait for her partner to act on the apparatus and maximize her rewards or access the secure alternative. In other words, the subject had some leverage over her partner (i.e. she could wait for her partner to access the unequal reward distribution under the certainty that the alternative was also available).

The experimenters found that chimpanzees did not use their leverage strategically, although they waited differently depending on the value of the alternative. The authors argued that the time delay of 10 s imposed between the subject and the partner having access to the apparatus could have increased the subjects' uncertainty about the possibility

of losing rewards, thus explaining why the subject did not wait for the partner to act. Furthermore, access to the alternative was more secure than access to the unequal reward distribution and allowed apes to avoid direct competition over food. In addition, subjects could use their leverage in half of the trials before the partner had accessed the apparatus, though in that situation the leverage could not influence the partners' decisions.

Given the constraints of the previous study, we returned to the UG to continue exploring whether chimpanzees and bonobos could use alternatives as a source of leverage to maximize gains. Specifically, we contrasted conditions in which the responder had leverage with normal UG conditions in which the partner could only decide between the proposers' offer or rejection based on inaction [16–19]. By contrast to the previous study by Sánchez-Amaro et al. [25], the responder always had to wait for the proposer to make an offer. This methodology facilitates the possibility for both individuals to pay attention to the alternative the partner has. Furthermore, even though the apes' choices are not simultaneous, we have removed any forced delay in between. Once the subject chooses the offer, the partner can decide whether to accept or reject it right afterwards.

Accordingly, we presented pairs of chimpanzees and bonobos with a mini UG where one proposer could choose between two offers. One offer was always beneficial to herself, while the other one benefitted the responder. Importantly, as in some of the previous UGs with great apes, we did not include the possibility to offer zero sums. Once the proposer made the offer, the responder could then decide whether to accept it or reject it. Both individuals obtained a reward if the responder accepted the offer. However, if the responder rejected the offer (e.g. by accessing the secure alternative), only the responder obtained a reward. We presented apes with three types of sessions: control sessions in which the proposers played alone and always obtained the offer they chose, test sessions with leverage for the responder and test sessions without leverage. Every test session consisted of two types of trials: trials with one of the two payoffs being clearly favourable for the proposer (FP) and trials with one of the two payoffs being clearly favourable for the responder (FR).

Proposers participated in four control sessions and both subjects participated in eight test sessions. There were four test sessions with leverage and four test sessions without leverage. In sessions with leverage, the responder had access to a positive alternative. In sessions without leverage, the responder had no access to a positive alternative. Every session consisted of FP and FR trials. In FP trials, the proposer could decide five rewards for the self and one for the responder or three rewards for each. In FR trials, the subject could decide one reward for the self and five for the responder or three rewards for each. For FP trials, the responder had access to a visible alternative of two rewards. In FR trials, the responder had access to a visible alternative of four rewards. This way, if proposers maximized in both conditions, responders could always obtain more rewards by choosing their alternative (figure 1).

We hypothesized that proposers would choose more favourable offers for the responder when compared to non-social control sessions. Furthermore, we hypothesized that proposers would be uncommonly willing to make fair offers for the responder when the latter had access to a positive alternative. Given the results from previous socio-cognitive

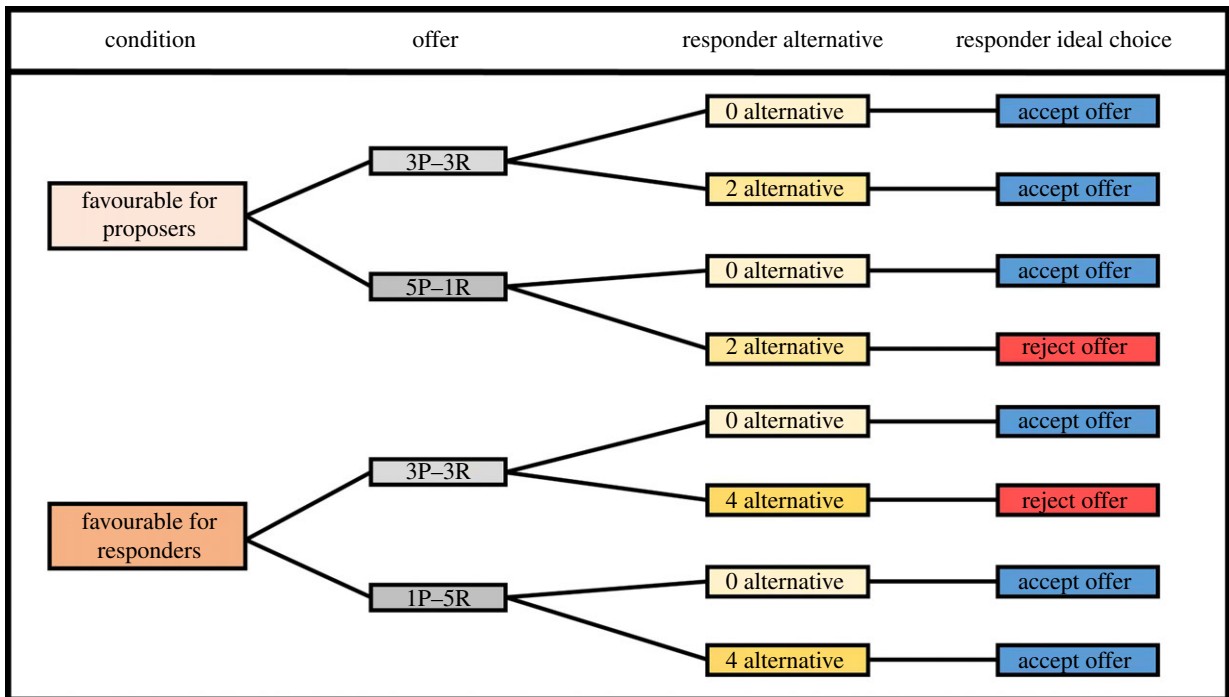

**Figure 1.** Representation of offers and alternatives for both conditions. P stands for proposer while R stands for responder. Responder ideal choice refers to the response that allows individuals to maximize their gain. (Online version in colour.)

studies, we hypothesize chimpanzees would be slightly more strategic than bonobos in their choices, given the higher social tolerance of bonobos and food-sharing tendencies [26–29] (but see [30–33] for alternative results suggesting no differences between species or even opposite results). That is, we would expect chimpanzees to be strategically prosocial when the responder has leverage, whereas we would expect bonobos to generally provide better offers to the responder regardless of the condition presented (although Kaiser *et al.* [18] did not find differences between bonobos and chimpanzees in their version of the UG). We also expected chimpanzee responders to behave as rational maximizers and access the visible alternative more often than bonobos, especially when the offer was less favourable than the alternative.

## 2. Material and methods

### (a) Participants

Twenty-four great apes (16 chimpanzees and eight bonobos) participated in at least one study stage. Nine great apes (six chimpanzees and three bonobos) participated in the final test phase. Apes were housed in Leipzig zoo (see electronic supplementary material, table S1, for more information about the apes that participated in the test phase). The nine individuals that participated in the test phase made up seven unique pairs of chimpanzees and three unique pairs of bonobos.

### (b) Materials

We presented individuals with a rectangular platform (78 × 33 cm) attached to a Plexiglas panel (73 × 64 cm) for the food preference test. We installed the panel on the front side of the apes' sleeping room. The platform could be slid back and forth between the Plexiglas panel and the experimenter. This movement facilitated both the baiting of the food by the experimenter and the apes' choices. The Plexiglas panel had two holes in the opposite bottom corners (3.2 cm in diameter). We used pellets for

chimpanzees and grapes for bonobos as food rewards. Food rewards were placed on white plastic dishes (7 cm in diameter). See electronic supplementary material, figures S1 and S2 for the apparatus representation.

We presented apes with a Plexiglas apparatus placed in a booth between two adjacent sleeping rooms for the pre-test and test sessions. The apparatus consisted of two sides, forming an L shape. The right side (proposer's side) consisted of two two-level compartments with a central gap between them, a ramp connected to the left side of the apparatus (responder's side), a Plexiglas door and an opening at the bottom. Each of the two-level compartments was composed of a top and a bottom tray connected. We baited each tray with different amounts of food.

When the ape slid the Plexiglas door to the left or the right side, they pushed one of the two-level compartments towards the centre of the apparatus. The food on the bottom tray of the compartment fell into the gap. The food on the top tray of the same compartment fell into the ramp and rolled down to the left side of the apparatus. The food located on the opposite two-level compartment (i.e. the one the proposer did not access) remained in the same position. Depending on the condition, the ape could then access the rewards that fell into the gap by inserting her hand through the bottom opening. However, on some occasions, a plastic lid inserted between the gap and the bottom opening blocked access to these food rewards. To detach the lid from the apparatus and make the rewards available, the same ape or another ape (depending on the condition, see below) had to operate the left side of the apparatus.

The left side consisted of a second Plexiglas door occluding two openings. The ape had to slide the door to the left to access the right-side opening (from the 'experimenters' perspective). By doing so, the ape could access the food that had previously rolled down from the right side of the apparatus (i.e. the food located on the top tray of the two-level compartment that the proposer had chosen). Furthermore, by sliding the door to the left, the ape could detach the plastic lid inserted between the gap and the bottom opening on the right side of the apparatus. Alternatively, the ape could slide the door to the right and access the left-side opening. This way, the responder could access an alternative

food compartment. In that case, the plastic lid was not detached from the apparatus.

On both sides of the apparatus, a locking mechanism prevented individuals from sliding the door more than once (i.e. they could not slide the door back to its initial position and access the other side).

## (c) Procedure
### (i) Preference test
We presented apes with a test to assess whether they would discern between the quantities involved in the test phase. All great apes participated in a minimum of two consecutive preference test sessions before they advanced to the first pre-test phase of the study. Each preference test session presented apes with eight choices between two different food quantities of banana-flavoured pellets, for chimpanzees, or grapes, for bonobos. Apes had to select the highest quantity in at least seven out of eight trials for two consecutive sessions. See more details in electronic supplementary material.

### (ii) First pre-test phase
Seventeen great apes participated in the first pre-test phase. Apes were individually tested in this and the following pre-test phase. In this phase, they had to learn how to operate both sides of the apparatus to obtain the rewards. First, the subject had to access one of the two-level compartments on the right side of the apparatus. The subject had to slide the Plexiglas door to the left or to the right to push the selected bottom tray towards the gap. The rewards fell on the plastic lid. Simultaneously, the rewards on the top tray of the selected compartment fell into the ramp and rolled down to the left side of the apparatus. The subject had to move to the left room and access the right-side opening on the left side of the apparatus. This action allowed the subject to access the rewards and to detach the lid blocking access to the rewards on the right side of the apparatus. Hence, after accessing the rewards on the left side of the apparatus, apes could go back to the right room to obtain the now available rewards through the bottom opening on the right side. The experimenter removed all other rewards placed on the apparatus (i.e. on the two-level compartment that they apes did not choose).

Apes experienced pre-test sessions consisting of eight trials. All trials presented apes with the decision between two and four rewards. One of the two two-level compartments was baited with two rewards on each tray, while the other compartment was baited with one reward on each tray. The side of the rewards varied pseudo-randomly within a session. Each quantity (two or four rewards) was presented on each of the two-level compartments in four trials. In addition, the same compartment could be baited with the same amount of food for a maximum of two consecutive trials within a session. Apes had to choose the compartment containing four rewards in at least seven out of eight trials on two consecutive sessions to advance to the second pre-test phase (see electronic supplementary material for more details).

### (iii) Second pre-test phase
Ten individuals participated in the second pre-test phase. In this phase, apes had to learn the distinction between the food presented on the left and right sides of the apparatus and maximize rewards. In this pre-test phase, only one of the two-level compartments was baited with one reward on each tray (two rewards in total). Apes had to select the compartment with rewards. The reward on the top tray fell into the ramp and rolled down to the left side of the apparatus. The reward on the bottom tray was pushed towards the gap and fell on the plastic lid. Apes had to manipulate the left side of the apparatus to access those rewards.

However, now the alternative food compartment contained either one or three food rewards. Apes had to slide the door to the right to access the alternative option through the left opening.

Great apes participated in pre-test sessions of eight trials. In half of the trials, the maximizing option was to access the two rewards baited on one of the two-level compartments (in these trials, the alternative food compartment was baited with a single reward). In the other half of the trials, the maximizing option was to access the alternative food compartment and obtain the three rewards. We pseudo-randomly counterbalanced the side of the baited trays and the number of rewards baited on the alternative food compartment. The same combination was not presented for more than two consecutive trials. The experimenter removed all the non-selected rewards once the ape made her choice. Apes had to maximize in at least seven out of eight trials on two consecutive sessions to advance to the test phase.

### (iv) Test phase
The 10 apes that participated in the second pre-test phase successfully advanced to the test phase. However, one chimpanzee could not participate in the test phase because no other group partner reached the criteria to advance to the test phase. Within a dyad, the individual that had access to the right side of the apparatus was the *proposer*, and the individual that had access to the left side of the apparatus was the *responder*.

Each dyad participated in ABA design with two types of sessions: control sessions and test sessions. The proposer participated in control and test sessions, while the responder only participated in test sessions. In control sessions, the proposer could only access the right side of the apparatus. In these sessions, the plastic lid did not prevent the proposer from obtaining the rewards from the selected bottom tray through the bottom opening. However, the proposer could not access the food baited on the selected top tray because the access to the left side was blocked (the food on the top tray always rolled down the ramp to the left side of the apparatus). Proposers participated in four control sessions, two before and two after the test sessions.

In test sessions, the proposer had access to the right side of the apparatus, while the responder had access to the left side of the apparatus for eight consecutive trials. The door separating both rooms remained closed. In test trials, the plastic lid was always inserted between the central hole and the bottom opening. Therefore, proposers could only access their selected rewards if responders had previously accessed the right-side opening on the left side. Apes participated in eight test sessions. In all but two dyads, individuals exchanged proposer–responder roles after the last control session.

In every session, the apes faced FP and FR conditions, which varied in the amount of food that the proposer and the responder could obtain. The order presentation of the conditions was pseudo-randomly counterbalanced: each condition was presented four times within a session, but never more than twice in a row, and they were equally represented on both compartments. In the FP condition, the proposer could decide between five food rewards for herself and one food reward for the responder (5–1 constellation) or three food rewards for each of them (3–3 constellation). The food rewards for the proposer were baited on the bottom trays of the two two-level compartments. The food rewards for the responder were baited on the top trays of the two two-level compartments. If, for example, the proposer chose the 5–1 constellation (instead of the 3–3 constellation), five food rewards were pushed towards the gap and one rolled down to the left side of the apparatus.

In the FR condition, the proposer could decide between one food reward for herself and five food rewards for the responder (1–5 constellation) or three food rewards for each dyad member (3–3 constellation). As in the previous condition, the food rewards for the proposer were baited on the bottom trays,

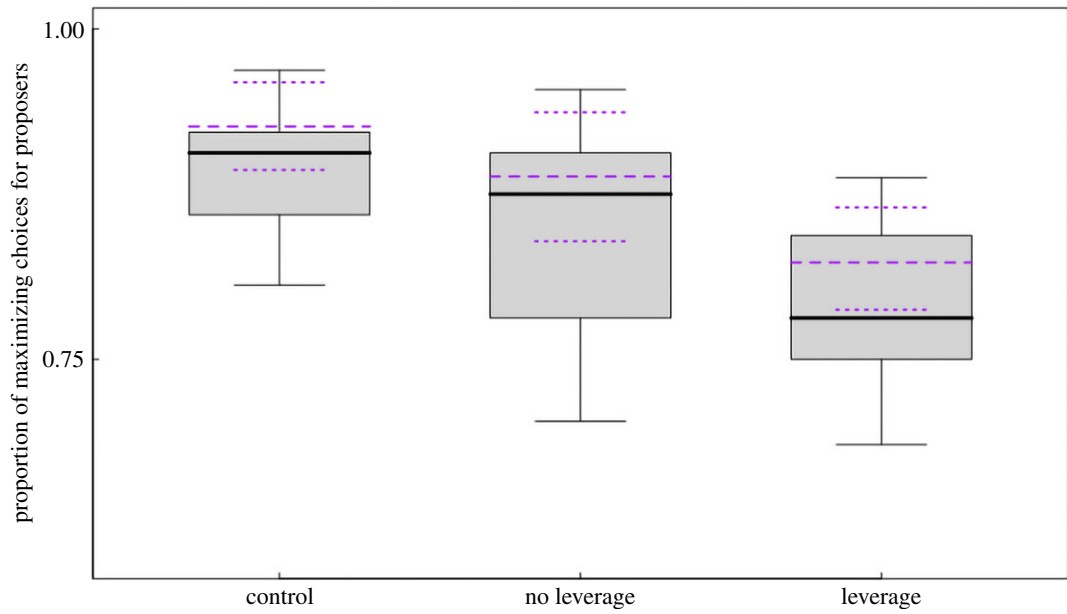

**Figure 2.** Proportion of trials where proposers maximized their choices as a function of the type of session (control, test session with no leverage and with leverage). Differences between conditions are significant. The box plot represents the median and q1 and q3 quartiles. The dotted lines in purple represented the fitted values of the model and the predicted 95% CI. (Online version in colour.)

and the food rewards for the responder were baited on the top trays.

The inclusion of favourable conditions for the proposer and the responder allowed us to control whether proposers would always prefer one specific food constellation. For instance, a proposer who would always choose the 3–3 constellation would only maximize her rewards in the FR condition. In addition, the inclusion of both conditions would let us detect any bias towards equality (e.g. favouring the 3–3 constellation across conditions) and prosociality (e.g. favouring the 1–5 constellation in the FR condition).

At the same time, in half of the test sessions, the responder had access to food located on the alternative food compartment (four test sessions with leverage and four test sessions without leverage). In sessions with leverage, the amount of food on the alternative compartment varied depending on the condition presented. During FP trials, the responder could access two rewards from the alternative platform. During FR trials, the responder could access four rewards from the alternative platform. In sessions with no leverage, there was no food baited on the alternative food compartment.

In sessions with leverage, if the responder accepted the proposer's offer, both individuals would obtain the reward constellation that the proposer had chosen. By contrast, when the responder preferred the food baited on the alternative compartment, the proposer would receive no rewards. Importantly, and regardless of the condition presented, the food rewards located on the alternative food compartment were always more significant than the rewards the responder would receive if the proposer decided to maximize, but lower than the food the responder would receive if the proposer decided not to maximize. Thus, the addition of leverage created opportunities for the responder to penalize the proposer's decisions. That is, the responder could reject the proposer's offer when that was not beneficial for the responder (table 1).

After the responder had decided, the experimenter removed all the rewards left on both sides of the apparatus. The two types of sessions (with and without leverage) alternated between test sessions, and the presentation order varied between pair members (e.g. one dyad member started the first test session as proposer with leverage and the other member started the first session as proposer without leverage).

## (d) Statistical analysis

All analyses were conducted with R statistics (v. 4.0.3). We used generalized linear mixed models (GLMMs) to investigate the decisions of subjects (model 1) and partners (models 2 and 3). Every full model was compared to a null model, excluding the test variables. In model 1, the test variables were species, type of session (three levels: control, test trials with leverage and test trials without leverage) and condition (two levels: FR and FP) as well as the two two-way interactions between species × type of session, and species × condition. In model 2, we included the same test variables without the interactions. Model 3 includes species, condition and whether the proposer had maximized or not as main effect variables. We controlled for session number, trial number and the experience of the proposer or responder in all our models. We included the ID of the proposer and the responder as random effects. See more details in electronic supplementary material.

## 3. Results

We assessed whether proposers considered the leverage of the responders when deciding between the two options. Interestingly, we found the main effect of type of session suggesting that both species chose more often the option that benefitted the responder when she had leverage, understood as access to the alternative option (GLMM: $\chi^2 = 9.49$, d.f. = 2, $p = 0.009$, $n = 1535$; figure 2). In other words, proposers maximized significantly less often during test sessions in which the responder could reject their offers. Pair-wise comparisons confirmed that the differences between the three types of sessions were significant. Proposers chose significantly more often the maximizing option for themselves in control sessions (when no responder was present) compared to test sessions in which the responder had no leverage ($p = 0.028$). At the same time, they chose the maximizing option significantly more often when the responder had no leverage than when the responder had leverage ($p = 0.033$)

Interestingly, we also found a two-way interaction between species and condition suggesting that bonobos maximized more than chimpanzees, especially during FR trials (GLMM: $\chi^2 = 4.32$, d.f. = 1, $p = 0.038$, $n = 1535$; electronic supplementary

**Figure 3.** Proportion of trials in which the responders accessed the alternative as a function of the 'proposers' decisions. Differences between species and between previous proposers decisions are significant. The box plot represents the median and q1 and q3 quartiles. The dotted lines in purple represented the fitted values of the model and the predicted 95% CI. (Online version in colour.)

material, figure S3). In addition, we found an effect of session suggesting that individuals maximized more by the end of the study. However, this result might be explained by the fact that the last two conditions were control conditions in which proposers should have maximized their gains. In fact, an updated model without including the control sessions confirmed no session effects (see electronic supplementary material, for further details of the models).

Not surprisingly, responder decisions were also affected by the leverage option in test sessions (apes never participated as responders in control sessions). Responders maximized significantly more often when they had no leverage available (GLMM: $\chi^2 = 14.02$, d.f. = 1, $p < 0.001$, $n = 1023$; electronic supplementary material, figure S4).

This result might seem contradictory at first sight, but it is not in the sense that when responders had leverage, they could decide whether to accept the proposer offer (even at a cost for themselves) or access the alternative. By contrast, when they had no leverage, they almost always accepted the proposers' initial offer—the alternative was to refuse the offer and, in consequence, obtain no food rewards. However, this result does not tell us whether responders used their leverage strategically. For that, we inspected whether the decisions of the responders to access their alternatives were influenced by the condition presented and by the proposers' previous decisions. We found a significant main effect of the previous proposer decision (GLMM: $\chi^2 = 19.52$, d.f. = 1, $p < 0.001$, $n = 512$; figure 3). Responders used their leverage more often when proposers maximized their choices. In other words, responders rejected more often offers below their alternative. Interestingly, we also found the main effect of species suggesting that chimpanzees exerted leverage over proposers more often than bonobos by accessing their alternative choice (GLMM: $\chi^2 = 4.85$, d.f. = 1, $p = 0.028$, $n = 512$; figure 3). Furthermore, there was variation between dyads with regards to the percentage of times in which proposers maximized offers and responders preferred the alternative when it was available (see electronic supplementary material, figure S5).

## 4. Discussion

In the current study, we presented chimpanzees and bonobos with a version of the UG in which responders could access alternatives as a source or bargaining leverage to reject the proposers' offers. Specifically, we investigated whether proposers would adjust their offers depending on the alternatives available for the responder and if responders would maximize their choices by using their leverage effectively—when proposers' offers were less favourable.

From the proposer perspective, we found that both species took into account the alternatives of the responder and adjusted their choices accordingly. That is, great apes were sensitive towards the responder's leverage and proposed fairer offers. A closer look at the results also revealed that, in general, chimpanzees seemed to act more strategically than bonobos to secure rewards in most trials: by selecting the least maximizing option, they could increase the chances to get their offer accepted. Nevertheless, we found no significant species differences in interaction with the type of session presented. We had initially hypothesized that chimpanzees would act more strategically than bonobos due to their more competitive nature [26,34]. However, while chimpanzees are less tolerant towards strangers and more aggressive in inter-community interactions [26,35,36], in food-sharing tasks, more akin to the context of this study, differences between bonobos and chimpanzees remain unclear [28,33]. Furthermore, the results are in line with a previous UG in which bonobos and chimpanzees behaved in similar ways [18].

We also found that bonobos proposers maximized slightly more than chimpanzees, especially during FR trials—and regardless of the type of session presented. They mainly chose the maximizing option for themselves, especially when they had to choose between one and three rewards. A closer look at this finding reveals that bonobos were primarily reactive towards one reward for the self (they maximized in 92–94% of trials across FR control and test sessions with and without leverage). However, that pattern was not present in FP trials. There, bonobos maximized more often in control compared to social sessions with and without leverage.

A possible reason bonobos maximized more from the proposer side than chimpanzees might relate to differences in motivation towards food. Perhaps the food we presented to the bonobos (grapes) was incredibly motivating for them—in contrast to how banana-flavoured pellets of similar size motivated chimpanzees. However, we doubt this was the case since we chose the different food types based on

previous research on food preferences in this population [37] and our discussion with ape keepers.

Another possibility might relate to differences in attention between species. Once the food was in place, bonobos could have restricted their attention to their side of the apparatus. This bias would have allowed them to maximize more than chimpanzees and to be more sensitive to changes in the ratio between quantities, possibly leading to higher rates of maximization in FR trials compared to FP trials (the difference in ratio between one and three is more prominent than between three and five). Furthermore, the same predisposition towards the rewards baited on the proposer side could have also prevented them from accurately computing the responders' options. This explanation also supports why bonobos did not differ between FR control and test trials.

Nonetheless, previous studies have found no chimpanzee–bonobo differences in quantity discrimination tasks with varying ratios [37,38]. Besides, all individuals participated in the same quantity preference task before the test. In that sense, we found no clear learning signature during training that would suggest differences between species in quantity discrimination and overall understanding of the task contingencies. Accordingly, we found no strong side-bias for any of our participants. The ape (a female chimpanzee) with the highest bias chose the left side of the proposer's apparatus in 58% of trials across conditions.

A third possibility is that bonobos could have been primarily reactive towards disadvantageous inequities [39,40]. Although they usually maximized their choices as proposers, they might have been less inclined to reject 3–3 offers in FP trials since both dyad members would obtain the same. By contrast, they might have been particularly aversive towards 1–5 offers in FR trials since that offer would increase inequality between the two pair members. This interpretation is in line with Bräuer et al. [41], who found that bonobos were the great ape species that showed the highest refusal rates in their inequity aversion task. Interestingly, this strategy could be supported by the fact that bonobos accepted more unequal offers than chimpanzees when playing the role of the responder. In other words, bonobos did not need to adjust their self-maximizing strategies as much as chimpanzees.

The reason why bonobos accepted more unequal offers than chimpanzees is intriguing. One possibility is that bonobos could have favoured conspecifics only when they could directly provide them with rewards: by accepting the offer, bonobos allowed both dyad members to obtain rewards, whereas they never benefitted the responder directly when they chose the offer as proposers. These results could have also been influenced by one of the three bonobos (the youngest female, Luiza) who was very willing to accept less favourable offers when she played as responder—although that difference in strategies was not apparent when she played the proposer role (electronic supplementary material, figure S2). Chimpanzees, in contrast, may have preferred to wait for the offer less than bonobos and thus accessed the alternative without considering the other option. Even though the offer was always released before the access to the alternative was unlocked, chimpanzees could have focused on the alternative from earlier on—the alternative was always closer to the responder. That could have explained their increased likelihood of accessing the alternative. However, note that chimpanzees still rejected the alternative most when the proposers' offer was favourable, meaning that they paid attention to both options.

Interestingly, the outcome of proposers' and responders' strategies resulted in more rewards for proposer bonobos compared to proposer chimpanzees, even though chimpanzee offers were more appealing for responders. At the same time, responder chimpanzees obtained slightly more rewards than responder bonobos by accessing the alternative option more often across conditions. Note here that although bonobos had to reach the same levels of accuracy as chimpanzees to advance through the pre-test phases, it is possible that they struggled to comprehend some of the task contingencies compared to chimpanzees. Although overall both species' offers were affected by the condition presented, bonobos did not really distinguish between control and test sessions in FP trials, and they used their leverage less often than chimpanzees. In addition, although we tested similar dyad numbers of both species, we were only able to test three unrelated bonobos (in contrast to a majority of kin-related dyads in chimpanzees; see electronic supplementary material, table S1). Thus, our bonobo sample size and potential kinship biases limit the interpretation of their behaviour in our task and might have enhanced the discrepancies we observe between the species performance.

Overall, great apes did not consistently play fair or prosocial. Instead, chimpanzees, and to lesser extent bonobos, adjusted their strategies across conditions depending on the responders' alternative. Furthermore, individuals understood the task dynamics from the beginning of the test trials and applied their strategies accordingly. Surprisingly, overall experience did not affect their strategies across time—although specific individuals behaved differently depending on their dyad partner (see correlation results; electronic supplementary material, figure S2).

The current study results differ from previous mini UGs in great apes. Crucially, the addition of positive alternatives as leverage for responders provided apes with opportunities to reject previous offers while still benefitting from the interaction. In the current study, apes no longer had to wait without acting to reject the offer (sometimes up to a minute [17,18]). Instead, rejections were clearer when the responder had access to the alternative. They chose one option or the other. In that sense, our study resembles more human UGs in which participants clearly state whether they want to accept or reject an original offer (e.g. [42]). The lack of alternatives in previous studies could also explain why apes most often accepted even zero offers. Possibly, apes preferred to accept offers given that there was no other option available and to move on to the next trial. Responses based on inaction appear to be hard for apes. It is also possible, as suggested by Smith & Silberberg [43], that apes accepted offers to maintain the rate of the reinforcement. Not surprisingly, when apes had no leverage, our results replicated previous findings: apes accepted the great majority of non-zero offers.

The leverage in the form of access to an alternative reward possibly provided responders with the bargaining power to affect the proposers' decisions, and proposers with the possibility to anticipate their actions. However, it is still unclear whether proposers anticipated potential rejections or whether previous rejections influenced proposers' future decisions. We favour the second interpretation since it is less cognitively demanding and does not require apes to infer their partners' future strategies.

Future studies should try to tease apart these two explanations. One possibility would be to present proposers with

scenarios in which the responder has already accessed the alternative versus scenarios in which the proposer can still decide whether to accept the offer or the alternative. In addition, the contingencies of our task constrained the apparatus design in ways that could have potentially increased the cognitive demands of the task. For example, proposers had to maintain up to five different food constellations in mind before offering the reward instead of the four constellations required in previous UGs (e.g. [16,17]). However, in contrast to previous UGs, we reduced the number of conditions to two (FP and FR). Finally, currently most of the studies highlighting differences between *Pan* species revolve around bonobos higher prosocial motivations for sharing and helping [28,44,45]. Nevertheless, bonobos and chimpanzees behaved quite similarly when a competitive component is present, such as in social dilemma games [18,46,47]. In that sense, future studies are necessary to continue shedding light on potential chimpanzee–bonobo differences in socio-cognitive tasks by directly contrasting individuals' prosocial, cooperative and competitive motivations across scenarios.

Our study improves previous attempts to understand whether great apes and children use social leverage to maximize benefits in social dilemmas [25,48]. By shortening the delay between proposers' and responders' decisions, we reduced potential uncertainty compared to previous studies. Also, we highlighted the causal connection between the usage of the leverage and the previous actions on the apparatus. In other words, only the offer being proposed could unlock the access to the alternative, whereas, in previous studies, the experimenter manipulated the apparatus following a schedule that was always unknown for the ape [25].

Overall, in our study, great apes still behaved as rational maximizers but in more strategic ways than previously stated. Proposers tended to behave strategically prosocial when responders had access to alternatives, possibly increasing their chances of benefiting from the interaction and responder maximized those alternatives accordingly.

Ethics. An internal ethics committee approved the study at the Max Planck Institute for Evolutionary Anthropology. The study complies with the Weatherfall report 'The use of non-human primates in research'. The study also complies with the EAZA Minimum Standards for the Accommodation and Care of Animals in Zoos and Aquaria, the WAZA Ethical Guidelines for the Conduct of Research on Animals by Zoos and Aquariums and the ASAB/ABS's Guidelines for the Treatment of Animals in Behavioural Research and Teaching. Chimpanzees and bonobos were housed in large semi-natural indoor and outdoor enclosures, and the research was conducted in their sleeping rooms. Great apes had regular feeding schedules, daily enrichments and water. Apes were never food- or water-deprived and could voluntarily participate in the test by entering their sleeping rooms. During the test sessions, great apes had access to water ad libitum.

Data accessibility. The data are provided in the electronic supplementary material [49].

Authors' contributions. A.S.-A.: Conceptualization, data curation, formal analysis, methodology, writing—original draft, writing—review and editing; F.R.: Conceptualization, methodology, resources, supervision, writing—review and editing

All authors gave final approval for publication and agreed to be held accountable for the work performed therein.

Competing interests. We declare we have no competing interests.

Funding. We received no funding for this study.

Acknowledgements. We thank the staff at the Wolfgang Kohler Primate Research Center in Leipzig Zoo for their support. We thank Linda Schymanski for the image representation of the task and Lara Schemion for reliability coding.

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
