## [Peer Review File · Proceedings of the Royal Society B: Biological Sciences]

Review History

RSPB-2021-1480.R0 (Original submission)

Review form: Reviewer 1

Recommendation

Major revision is needed (please make suggestions in comments)

Scientific importance: Is the manuscript an original and important contribution to its field?

Excellent

General interest: Is the paper of sufficient general interest?

Excellent

Quality of the paper: Is the overall quality of the paper suitable?

Good

Is the length of the paper justified?

No

Should the paper be seen by a specialist statistical reviewer?

No

Do you have any concerns about statistical analyses in this paper? If so, please specify them explicitly in your report.

No

It is a condition of publication that authors make their supporting data, code and materials available - either as supplementary material or hosted in an external repository. Please rate, if applicable, the supporting data on the following criteria.

Is it accessible?

Yes

Is it clear?

Yes

Is it adequate?

Yes

Do you have any ethical concerns with this paper?

No

Comments to the Author

This study presents an experiment chimpanzees and bonobos as subjects. It uses a variant of a mini ultimatum game, in which the responder gets an outside option if she rejects. The outside option is chosen such that if the proposer chooses the kind option, it in the material self-interest of the responder to accept, and if the proposer chooses the unkind option, it in the material self-interest of the responder to reject. The results show that both, chimpanzees as well as bonobos, strongly discriminate between the kind and the unkind offer and reject the unkind offer frequently. The bonobos choose the rejection a bit less likely, which could be interpreted that the bonobo responders are somewhat kinder. The chimpanzee proposers anticipate this behavior and choose the kind offer with a higher probability, while this is not much less the case with the bonobos.

The design is very interesting. The responder behavior shows whether they take the outside option into account. They do it to a high degree. This behavior is not so surprising because the alternatives are directly visible. The proposers have a more difficult task. They have to anticipate the behavior of the responder, and they have to determine the expected benefit of the two alternatives. Because the rejection rates are rather high, it would be expected payoff maximizing (and less risky) to choose the kind option if the outside option is as in the experiment. It is surprising that this is not the case, which means that they take the behavior of the responders only incompletely into account. Further, the treatments vary in whether the kind or the unkind option is fair. Different to human subjects, the apes do not make much of a difference.

The experiment is very well set up and contains good control conditions. The statistical analysis and the presentation of the result is clear. However, the motivation does not convince me. The fact that the apes take the "possibility of using social leverage" means only that they take the alternative into account – and this alternative is clearly visible. In my view the study addresses an interesting topic but I disagree with the authors about what is interesting. In my view, the change in the responder behavior is not particularly surprising, also in comparison to the ultimatum game. Nevertheless, the experiment is an interesting setup in order to investigate anticipation of the behavior of others.

I recommend refocusing the paper. First, I recommend explaining the experiment in simpler terms. I would present Table 1 earlier and I would (also) present it in a game tree. Further, the terms "selfish condition" and "prosocial condition" are confusing. The conditions are not selfish; only a decision can be selfish. The conditions differ in whether the options favor the proposer or the responder. Second, I would develop hypothesis based on the game theoretical predictions. According to this prediction, the responder behavior is not so surprising. In the last paragraph, it is written that "great apes still behaved as rational maximizers but in more nuanced strategic ways than previously stated. For example, responders did not accept any non-zero offer." I do

not see in what respect the behavior is nuanced. The responders usually just took the selfishly better option. But as stated above, the experiment provides an interesting setup to study how the strategic situation is taken into account.

Refocusing the paper could also reduce it somewhat in length.

Review form: Reviewer 2 (Nicky Staes)

Recommendation

Major revision is needed (please make suggestions in comments)

Scientific importance: Is the manuscript an original and important contribution to its field?

Good

General interest: Is the paper of sufficient general interest?

Good

Quality of the paper: Is the overall quality of the paper suitable?

Acceptable

Is the length of the paper justified?

Yes

Should the paper be seen by a specialist statistical reviewer?

No

Do you have any concerns about statistical analyses in this paper? If so, please specify them explicitly in your report.

Yes

It is a condition of publication that authors make their supporting data, code and materials available - either as supplementary material or hosted in an external repository. Please rate, if applicable, the supporting data on the following criteria.

Is it accessible?

Yes

Is it clear?

Yes

Is it adequate?

Yes

Do you have any ethical concerns with this paper?

No

Comments to the Author

The authors present an impressive experimental version of the ultimatum game performed by chimpanzees and bonobos to test whether they use social leverage strategically in a feeding context. They show that both species maximize their food reward depending on the presence of leverage. They also find that bonobos overall maximize more than chimpanzees and that there were no differences between the species in patterns of tests or between conditions.

While we agree with the authors that the patterns is likely robust in chimpanzees, we question to some extent the results in the bonobos and therefore the interpretation of the results in light of

species differences. Looking at bonobo maximizing behavior across conditions, it looks like they maximize more mainly in the prosocial condition vs chimpanzees, and not so much in the selfish condition. It also seems that within the prosocial condition, they maximize more in the social condition with or without leverage than they do in the control, which would indicate that they may not understand the set-up very well or that there was potentially some sort of testing bias present. These concerns are explained in more detail below. While the small sample size in bonobos might not allow for three-way interaction effects between species, leverage and conditions, some discussion at the least is needed on this.

Abstract and introduction

Line 41: change “we offer chimpanzees and bonobos responders the possibility” to “we offer chimpanzee and bonobo responders the possibility”

Line 97: but see also:

Oberliessen & Kalenscher, 2019 Social and non-social mechanisms of inequity aversion in non-human animals. *Front. Behav. Neurosci.*

Line 108: I'd recommend rephrasing this sentence. It's not clear from reading this what you mean with “visual cues of estrous over extended periods” give females leverage. I understand that ovulation is hidden in bonobo females because they have a longer period of sexual swelling, and that this is a mechanism that ultimately potentially protects them against male mate guarding but this is not really clear from this sentence. I'm also not sure this is a great example of social leverage tactics since it's not really something the females actively decide to do but rather a mechanism that developed through years of natural/sexual selection.

Line 113 & 141: In the light of the ability to “consciously” use the leverage: Did you observe any begging, harassment behavior or other communication from the responder before the choice by the proposer that may have influenced the proposer's choice? To what extent did they pay attention to each other or to the choices they made? Did you observe any reaction after the responder received the offer?

Line 174: The view that bonobos are more tolerant than chimpanzees is a bit outdated and dependent on what is considered as a measure for tolerance. It is true that they have higher levels of social tolerance in the form of intercommunity interactions (reduced rates of aggression and lethal aggression) but when it comes to feeding tolerance, there is a lot of inconsistency in the outcomes of different experiments. The studies cited are indeed examples of tolerant food sharing but there is plenty of additional literature claiming the reverse or at least equal feeding tolerance and large within-species variation that exceeds between-species variation. This is also relevant for your discussion where you claim that bonobos are less competitive around food and more willing to share. For example see:

Amici, F., Call, J. & Aureli, F. Aversion to violation of expectations of food distribution: The role of social tolerance and relative dominance in seven primate species. *Behaviour* 149, 345–368 (2012)

Bullinger, A. F., Burkart, J. M., Melis, A. P. & Tomasello, M. Bonobos, *Pan paniscus*, chimpanzees, *Pan troglodytes*, and marmosets, *Callithrix jacchus*, prefer to feed alone. *Anim. Behav.* 85, 51–60 (2013)

Cronin, K. A., De Groot, E. & Stevens, J. M. G. Bonobos show limited social tolerance in a group setting: A comparison with chimpanzees and a test of the relational model. *Folia Primatol.* 86, 164–177 (2015).

Jaeggi, A. V., De Groot, E., Stevens, J. M. G. & Van Schaik, C. P. Mechanisms of reciprocity in primates: Testing for short-term contingency of grooming and food sharing in bonobos and chimpanzees. *Evol. Hum. Behav.* 34, 69–77 (2013).

Jaeggi, A. V., Stevens, J. M. G. & Van Schaik, C. P. Tolerant food sharing and reciprocity is precluded by despotism among bonobos but not chimpanzees. *Am. J. Phys. Anthropol.* 143, 41–51 (2010).

Methods

Line 187-192: I'd add some information about relatedness for the remaining dyads, I don't think it's an issue here but it's good to add, especially given the small sample size and the important role kinship could play on food sharing.

Line 249-260: To me the figures are a bit confusing, likely because of the complex set-up. Is there a way to make the rewards stand out more? I realize they are already a bright green but it's still a

bit hard to tell.

Line 267-270: Can you add here why you did this test? What came out of the preference tests? When was decided whether subjects were able to discriminate between quantities? Did they need to reach a specific criterium?

L282 Double space between "... apparatus. Hence..."

L297 What was the criterium that subjects had to pass before proceeding to the next test phase?

L317: "Ten apes... successfully advanced to the test phase" but what does this mean? The validity of the outcome of this study depends on this phase of the pre-test training. Therefore, a more detailed description of this part of the study is necessary.

Line 390: typo, change "." to "," after session. Also, you tested for species effects but there is no mention of this in your statistics section in the main manuscript. Did you test for species by condition/test interaction effects? This would be key to interpret your results.

Results and discussion

Can you show in the figures which groups differed significantly from others? Either below in the text or in the figure itself using asterisks. I think figure 2 is a bit distracting since it does not show the results of your analysis: neither the species effect, which lumps the data from all leverage levels together, or leverage effect, which lumps data from both species together.

Line 457: It's unclear why your results are in line with higher feeding tolerance in bonobos. Isn't the fact that they always maximize more in line with lower feeding tolerance? You do mention later that they do not necessarily behave more prosocially but maybe the fact that they do not distinguish between the test conditions means they did not really understand?

Did you consider analyzing a species by condition interaction effect? When looking at the proportion of maximized choices across conditions and species, the higher levels in bonobos compared to chimpanzees seem to be present primarily in the prosocial condition in test trials, but not control trials: bonobos seem to maximize less in the prosocial condition in the absence of a responder. This is surprising if bonobos are expected to take others behavior into account. This contrasts with line 398-404. Moreover, if bonobos prefer 3 over 1 grape, why do they show such low maximizing levels in the prosocial trials in the control sessions, not showing the self-maximizing offers (Line 468)? Why would a subject choose the '1-5' option in the prosocial control condition at all? (The same is true for the '3-3' option in the selfish trials in the control condition, which is even lower for bonobos). The figure that we attached to this document is based on the data provided by the authors with the manuscript and reflects this issue more clearly.

Line 477 "Bonobos accepted more unequal offers" and line 498 "maximized more regardless of the responders options". In combination with the previous comment, this raises the question whether bonobos did understand task contingencies. Even though no bias towards a specific constellation across conditions was found, other aspects could have caused the relatively small differences in outcome (like task complexity, local enhancement due to presence of the conspecific on the right side). Did you test whether any side-biases were present in the actual test phase of the experiment? e.g did subjects that had to be retested in the first pre-test phase show higher or lower maximizing scores or biases? Or did (some of) the individuals show a slight left-sliding preference (cfr lower choices for alternative offer)?

What did the set-up look like on the left-side in the no-leverage condition? Although there was no food baited on the alternative food compartment, did the responders still have to slide the door to the left to access the offer of the proposer? Could there be any carry-over effect of these no-leverage sessions on the leverage-sessions that may have resulted in lower overall alternative choices by bonobos? Or did bonobos prefer to keep their distance from the conspecific resulting in a preference for the right sight of the apparatus, i.e. the side of the offer by the proposer?

Line 486-488: This is in contrast to what was previously said about bonobos being risk averse and chimpanzees choosing the riskier option, in line with differences in their feeding ecology. Maybe this deserves some discussion here as well:

Heilbronner, S.R., Rosati, A.G., Stevens, J.R., Hare, B., Hauser, M.D. (2008). A fruit in the hand or two in the bush? Divergent risk preferences in chimpanzees and bonobos. *Biology Letters*, -1(-1), -1-1. DOI: 10.1098/rsbl.2008.0081

Line 523-524: How does the amount of maximization differ between the control trials in sessions

1-2 and 11-12? The data do suggest an increase in maximization, hence possible task comprehension?

Supplement

Figure S2: Squares should represent bonobo-dyads, which is not true for Yasa-Kuno: represented by a pink circle in stead of a pink square; Sandra-Frodo, a chimpanzee dyad, should be represented by a circle.

Data

check data lines

- 129: selfish condition control- choice subject 1: corresponds with prosocial condition?
- 864: prosocial condition - choice subject 5, choice partner 2 corresponds with selfish condition
- 392; 739; 1125: Alternative column is empty

How did you take trials into account where subjects and partners made a choice but subjects did not eat the food. It seems that especially for the dyad Kuno-Yasa this happened 8 times. Did you consider these trials in your analyses? What would these choices mean e.g choosing the maximized option in the selfish condition with a partner that agreed with the offer but then the subject decided to not consume the food items? What happens if you exclude them from the analyses?

Please check whether these suggestions concerning the data influence analyses and results.

Decision letter (RSPB-2021-1480.R0)

10-Aug-2021

Dear Dr Sánchez Amaro:

I have now received comments from two reviewers and the Associate Editor, and am writing to inform you that your manuscript RSPB-2021-1480 entitled "Chimpanzees and bonobos use social leverage in an Ultimatum Game" has, in its current form, been rejected for publication in Proceedings B. This action has been taken on the advice of referees, who have recommended that substantial revisions are necessary. However, based on my read of your manuscript as well as the reviewers' and AE's comments, I would be happy to consider a resubmission, provided the comments of the referees are fully addressed. In particular, please pay attention to reviewer 2's comments regarding the analyses and interpretation. With the caveat that I am, as you know, involved in this literature and have my own thoughts on it, I would add to their comments that the discussion of previous work on the UG in chimpanzees could be a bit more nuanced, especially given that your results are in line with the research suggesting that chimpanzees do sometimes change their behavior in UGs and this concordance might contribute to understanding what circumstances lead to this reaction. However, I leave it to you whether and how you wish to address this point. Finally, I agree with reviewer 1 that the manuscript could be simplified which would make it much more accessible to those outside the field. Of course, this is not a provisional acceptance and we will send your revised manuscript back out for review before making a final decision.

Sincerely,
 Dr Sarah Brosnan
 Editor, Proceedings B
 mailto: proceedingsb@royalsociety.org

Associate Editor
 Board Member: 1
 Comments to Author:

This manuscript was reviewed by two experts and myself. We all see many positives of the study and the use of novel and comparative cognitive experiments in chimpanzees and bonobos. The goal of understanding how these animals account for the options available to their partner in versions of the ultimatum game is important and has direct relevance to understanding human cognitive evolution. However, there are problems that need to be addressed before the manuscript might be suitable for publication in Proceedings B. Reviewer 1 has several helpful suggestions for reframing and streamlining the manuscript to make the experimental approach and interpretation clearer for the reader. Reviewer 2 notes some inconsistencies with the data and presentation that need to be clarified. In addition, Reviewer 2 notes several areas for clarification and improvement, including providing more details about the experiments and new ways to analyze the data and present the results. Together these reviewer suggestions amount to streamlining the framing but providing more details in the methods and results as well as new ways to analyze and present the data. Reviewer 2 also notes issues with sample size in bonobos which needs to be acknowledged in the Discussion. If these issues can be addressed, I would consider a revised version which will go out for further review, hopefully to the same reviewers.

Reviewer(s)' Comments to Author:

Referee: 1

Comments to the Author(s)

This study presents an experiment chimpanzees and bonobos as subjects. It uses a variant of a mini ultimatum game, in which the responder gets an outside option if she rejects. The outside option is chosen such that if the proposer chooses the kind option, it is in the material self-interest of the responder to accept, and if the proposer chooses the unkind option, it is in the material self-interest of the responder to reject. The results show that both, chimpanzees as well as bonobos, strongly discriminate between the kind and the unkind offer and reject the unkind offer frequently. The bonobos choose the rejection a bit less likely, which could be interpreted that the bonobo responders are somewhat kinder. The chimpanzee proposers anticipate this behavior and choose the kind offer with a higher probability, while this is not much less the case with the bonobos.

The design is very interesting. The responder behavior shows whether they take the outside option into account. They do it to a high degree. This behavior is not so surprising because the alternatives are directly visible. The proposers have a more difficult task. They have to anticipate

the behavior of the responder, and they have to determine the expected benefit of the two alternatives. Because the rejection rates are rather high, it would be expected payoff maximizing (and less risky) to choose the kind option if the outside option is as in the experiment. It is surprising that this is not the case, which means that they take the behavior of the responders only incompletely into account. Further, the treatments vary in whether the kind or the unkind option is fair. Different to human subjects, the apes do not make much of a difference.

The experiment is very well set up and contains good control conditions. The statistical analysis and the presentation of the result is clear. However, the motivation does not convince me. The fact that the apes take the “possibility of using social leverage” means only that they take the alternative into account – and this alternative is clearly visible. In my view the study addresses an interesting topic but I disagree with the authors about what is interesting. In my view, the change in the responder behavior is not particularly surprising, also in comparison to the ultimatum game. Nevertheless, the experiment is an interesting setup in order to investigate anticipation of the behavior of others.

I recommend refocusing the paper. First, I recommend explaining the experiment in simpler terms. I would present Table 1 earlier and I would (also) present it in a game tree. Further, the terms “selfish condition” and “prosocial condition” are confusing. The conditions are not selfish; only a decision can be selfish. The conditions differ in whether the options favor the proposer or the responder. Second, I would develop hypothesis based on the game theoretical predictions. According to this prediction, the responder behavior is not so surprising. In the last paragraph, it is written that “great apes still behaved as rational maximizers but in more nuanced strategic ways than previously stated. For example, responders did not accept any non-zero offer.” I do not see in what respect the behavior is nuanced. The responders usually just took the selfishly better option. But as stated above, the experiment provides an interesting setup to study how the strategic situation is taken into account.

Refocusing the paper could also reduce it somewhat in length.

Referee: 2

Comments to the Author(s)

The authors present an impressive experimental version of the ultimatum game performed by chimpanzees and bonobos to test whether they use social leverage strategically in a feeding context. They show that both species maximize their food reward depending on the presence of leverage. They also find that bonobos overall maximize more than chimpanzees and that there were no differences between the species in patterns of tests or between conditions.

While we agree with the authors that the patterns is likely robust in chimpanzees, we question to some extent the results in the bonobos and therefore the interpretation of the results in light of species differences. Looking at bonobo maximizing behavior across conditions, it looks like they maximize more mainly in the prosocial condition vs chimpanzees, and not so much in the selfish condition. It also seems that within the prosocial condition, they maximize more in the social condition with or without leverage than they do in the control, which would indicate that they may not understand the set-up very well or that there was potentially some sort of testing bias present. These concerns are explained in more detail below. While the small sample size in bonobos might not allow for three-way interaction effects between species, leverage and conditions, some discussion at the least is needed on this.

Abstract and introduction

Line 41: change “we offer chimpanzees and bonobos responders the possibility” to “we offer chimpanzee and bonobo responders the possibility”

Line 97: but see also:

Oberliessen & Kalenscher, 2019 Social and non-social mechanisms of inequity aversion in non-human animals. *Front. Behav. Neurosci.*

Line 108: I'd recommend rephrasing this sentence. It's not clear from reading this what you mean with “visual cues of estrous over extended periods” give females leverage. I understand that ovulation is hidden in bonobo females because they have a longer period of sexual swelling, and that this is a mechanism that ultimately potentially protects them against male mate guarding but this is not really clear from this sentence. I'm also not sure this is a great example of social

leverage tactics since it's not really something the females actively decide to do but rather a mechanism that developed through years of natural/sexual selection.

Line 113 & 141: In the light of the ability to "consciously" use the leverage: Did you observe any begging, harassment behavior or other communication from the responder before the choice by the proposer that may have influenced the proposer's choice? To what extent did they pay attention to each other or to the choices they made? Did you observe any reaction after the responder received the offer?

Line 174: The view that bonobos are more tolerant than chimpanzees is a bit outdated and dependent on what is considered as a measure for tolerance. It is true that they have higher levels of social tolerance in the form of intercommunity interactions (reduced rates of aggression and lethal aggression) but when it comes to feeding tolerance, there is a lot of inconsistency in the outcomes of different experiments. The studies cited are indeed examples of tolerant food sharing but there is plenty of additional literature claiming the reverse or at least equal feeding tolerance and large within-species variation that exceeds between-species variation. This is also relevant for your discussion where you claim that bonobos are less competitive around food and more willing to share. For example see:

Amici, F., Call, J. & Aureli, F. Aversion to violation of expectations of food distribution: The role of social tolerance and relative dominance in seven primate species. *Behaviour* 149, 345-368 (2012)

Bullinger, A. F., Burkart, J. M., Melis, A. P. & Tomasello, M. Bonobos, *Pan paniscus*, chimpanzees, *Pan troglodytes*, and marmosets, *Callithrix jacchus*, prefer to feed alone. *Anim. Behav.* 85, 51-60 (2013)

Cronin, K. A., De Groot, E. & Stevens, J. M. G. Bonobos show limited social tolerance in a group setting: A comparison with chimpanzees and a test of the relational model. *Folia Primatol.* 86, 164-177 (2015).

Jaeggi, A. V., De Groot, E., Stevens, J. M. G. & Van Schaik, C. P. Mechanisms of reciprocity in primates: Testing for short-term contingency of grooming and food sharing in bonobos and chimpanzees. *Evol. Hum. Behav.* 34, 69-77 (2013).

Jaeggi, A. V., Stevens, J. M. G. & Van Schaik, C. P. Tolerant food sharing and reciprocity is precluded by despotism among bonobos but not chimpanzees. *Am. J. Phys. Anthropol.* 143, 41-51 (2010).

Methods

Line 187-192: I'd add some information about relatedness for the remaining dyads, I don't think it's an issue here but it's good to add, especially given the small sample size and the important role kinship could play on food sharing.

Line 249-260: To me the figures are a bit confusing, likely because of the complex set-up. Is there a way to make the rewards stand out more? I realize they are already a bright green but it's still a bit hard to tell.

Line 267-270: Can you add here why you did this test? What came out of the preference tests? When was decided whether subjects were able to discriminate between quantities? Did they need to reach a specific criterium?

L282 Double space between "... apparatus. Hence..."

L297 What was the criterium that subjects had to pass before proceeding to the next test phase?

L317: "Ten apes... successfully advanced to the test phase" but what does this mean? The validity of the outcome of this study depends on this phase of the pre-test training. Therefore, a more detailed description of this part of the study is necessary.

Line 390: typo, change "." to "," after session. Also, you tested for species effects but there is no mention of this in your statistics section in the main manuscript. Did you test for species by condition/test interaction effects? This would be key to interpret your results.

Results and discussion

Can you show in the figures which groups differed significantly from others? Either below in the text or in the figure itself using asterisks. I think figure 2 is a bit distracting since it does not show the results of your analysis: neither the species effect, which lumps the data from all leverage levels together, or leverage effect, which lumps data from both species together.

Line 457: It's unclear why your results are in line with higher feeding tolerance in bonobos. Isn't the fact that they always maximize more in line with lower feeding tolerance? You do mention later that they do not necessarily behave more prosocially but maybe the fact that they do not distinguish between the test conditions means they did not really understand?

Did you consider analyzing a species by condition interaction effect? When looking at the proportion of maximized choices across conditions and species, the higher levels in bonobos compared to chimpanzees seem to be present primarily in the prosocial condition in test trials, but not control trials: bonobos seem to maximize less in the prosocial condition in the absence of a responder. This is surprising if bonobos are expected to take others behavior into account. This contrasts with line 398-404. Moreover, if bonobos prefer 3 over 1 grape, why do they show such low maximizing levels in the prosocial trials in the control sessions, not showing the self-maximizing offers (Line 468)? Why would a subject choose the '1-5' option in the prosocial control condition at all? (The same is true for the '3-3' option in the selfish trials in the control condition, which is even lower for bonobos). The figure that we attached to this document is based on the data provided by the authors with the manuscript and reflects this issue more clearly.

Line 477 "Bonobos accepted more unequal offers" and line 498 "maximized more regardless of the responders options". In combination with the previous comment, this raises the question whether bonobos did understand task contingencies. Even though no bias towards a specific constellation across conditions was found, other aspects could have caused the relatively small differences in outcome (like task complexity, local enhancement due to presence of the conspecific on the right side). Did you test whether any side-biases were present in the actual test phase of the experiment? e.g did subjects that had to be retested in the first pre-test phase show higher or lower maximizing scores or biases? Or did (some of) the individuals show a slight left-sliding preference (cfr lower choices for alternative offer)?

What did the set-up look like on the left-side in the no-leverage condition? Although there was no food baited on the alternative food compartment, did the responders still have to slide the door to the left to access the offer of the proposer? Could there be any carry-over effect of these no-leverage sessions on the leverage-sessions that may have resulted in lower overall alternative choices by bonobos? Or did bonobos prefer to keep their distance from the conspecific resulting in a preference for the right side of the apparatus, i.e. the side of the offer by the proposer?

Line 486-488: This is in contrast to what was previously said about bonobos being risk averse and chimpanzees choosing the riskier option, in line with differences in their feeding ecology. Maybe this deserves some discussion here as well:

Heilbronner, S.R., Rosati, A.G., Stevens, J.R., Hare, B., Hauser, M.D. (2008). A fruit in the hand or two in the bush? Divergent risk preferences in chimpanzees and bonobos. *Biology Letters*, 1(1), 1-1. DOI: 10.1098/rsbl.2008.0081

Line 523-524: How does the amount of maximization differ between the control trials in sessions 1-2 and 11-12? The data do suggest an increase in maximization, hence possible task comprehension?

Supplement

Figure S2: Squares should represent bonobo-dyads, which is not true for Yasa-Kuno: represented by a pink circle in stead of a pink square; Sandra-Frodo, a chimpanzee dyad, should be represented by a circle.

Data

check data lines

- 129: selfish condition control- choice subject 1: corresponds with prosocial condition?
- 864: prosocial condition - choice subject 5, choice partner 2 corresponds with selfish condition
- 392; 739; 1125: Alternative column is empty

How did you take trials into account where subjects and partners made a choice but subjects did not eat the food. It seems that especially for the dyad Kuno-Yasa this happened 8 times. Did you consider these trials in your analyses? What would these choices mean e.g choosing the maximized option in the selfish condition with a partner that agreed with the offer but then the subject decided to not consume the food items? What happens if you exclude them from the analyses?

Please check whether these suggestions concerning the data influence analyses and results.

Author's Response to Decision Letter for (RSPB-2021-1480.R0)

See Appendix A.

RSPB-2021-1937.R0

Review form: Reviewer 1

Recommendation

Accept with minor revision (please list in comments)

Scientific importance: Is the manuscript an original and important contribution to its field?

Excellent

General interest: Is the paper of sufficient general interest?

Excellent

Quality of the paper: Is the overall quality of the paper suitable?

Excellent

Is the length of the paper justified?

Yes

Should the paper be seen by a specialist statistical reviewer?

No

Do you have any concerns about statistical analyses in this paper? If so, please specify them explicitly in your report.

No

It is a condition of publication that authors make their supporting data, code and materials available - either as supplementary material or hosted in an external repository. Please rate, if applicable, the supporting data on the following criteria.

Is it accessible?

Yes

Is it clear?

Yes

Is it adequate?

Yes

Do you have any ethical concerns with this paper?

No

Comments to the Author

This revision satisfactorily addresses most of my issues of the previous version. I also like the additional analysis. There is one remaining point that I recommend to address: While I agree that the proposers (can) act strategically, I do not see this possibility for the responders – at least not in the form as it is used. I outline the problem at two specific text passages.

Abstract, last sentence “Our results suggest that great apes act as rational maximizers in an Ultimatum Game, yet can also act strategically to maximize their payoff depending on their role (proposer vs. responder) and access to alternatives.” First, strategic behavior is not in contradiction or complementing rational behavior. It is rational to take the behavior of other into account. Second, the responder needs no strategic considerations to make. The responder sees what the proposer has done and can respond. Strategic behavior on the side of the responder would only be relevant if the responder takes future interaction into account; for example, caring about a reputation for being tough. Such behavior seems not to be present in the data.

Page 3, “Furthermore, UGs can complement observational studies [15] to help us investigate whether non-human primates use leverage strategically.” It is unclear what “strategic use” of the leverage means. The responders are simply better off if they reject in the case of leverage. My understanding of “strategic use” means that one gives up an opportunity for higher gains in the future. For example, rejection in the UG can be strategic if the game is repeated.

Typo

Page 5: “payofs” in “trials with one of the two payofs being clearly favorable for the responder”

Review form: Reviewer 2

Recommendation

Accept with minor revision (please list in comments)

Scientific importance: Is the manuscript an original and important contribution to its field?

Good

General interest: Is the paper of sufficient general interest?

Good

Quality of the paper: Is the overall quality of the paper suitable?

Good

Is the length of the paper justified?

Yes

Should the paper be seen by a specialist statistical reviewer?

No

Do you have any concerns about statistical analyses in this paper? If so, please specify them explicitly in your report.

No

It is a condition of publication that authors make their supporting data, code and materials available - either as supplementary material or hosted in an external repository. Please rate, if applicable, the supporting data on the following criteria.

Is it accessible?

Yes

Is it clear?

Yes

Is it adequate?

Yes

Do you have any ethical concerns with this paper?

No

Comments to the Author

We are happy with the revisions made by the authors with the exception of one minor adjustment that we would recommend needs to be made to increase transparency. As we had asked to include information about genetic relatedness of the dyads, the table in the supplement now reveals that kinship might actually be an issue causing a bias in the sample. While almost all chimpanzee dyads were related, none of the bonobo dyads were. The potential effect of kinship on foodsharing should thus be reviewed in the discussion and mentioned as a potential bias in the cross-species comparison.

Decision letter (RSPB-2021-1937.R0)

06-Oct-2021

Dear Dr Sánchez Amaro

I am very pleased to inform you that your manuscript RSPB-2021-1937 entitled "Chimpanzees and bonobos use social leverage in an Ultimatum Game" has been accepted for publication in Proceedings B. Personally, I really like your study and think that your paper will be a strong contribution to the literature. Please note that the reviewers have recommended a few minor revisions to improve the clarity of your manuscript (see their comments, appended below). In addition, I wonder if in Figure 1 it might clarify to indicate that the "Responder Choice" is really their ideal choice to maximize their outcome, not what they picked in the study (and I agree with the reviewers that the inclusion of the figure makes it much easier to follow the method for those who won't be as familiar with the general paradigm, so thank you for doing so). I invite you to respond to the referee(s)' comments and revise your manuscript. Because the schedule for publication is very tight, it is a condition of publication that you submit the revised version of your manuscript within 7 days. If you do not think you will be able to meet this date please let us know.

1) A text file of the manuscript (doc, txt, rtf or tex), including the references, tables (including captions) and figure captions. Please remove any tracked changes from the text before submission. PDF files are not an accepted format for the "Main Document".

2) A separate electronic file of each figure (tiff, EPS or print-quality PDF preferred). The format should be produced directly from original creation package, or original software format. PowerPoint files are not accepted.

3) Electronic supplementary material: this should be contained in a separate file and where possible, all ESM should be combined into a single file. All supplementary materials accompanying an accepted article will be treated as in their final form. They will be published alongside the paper on the journal website and posted on the online figshare repository. Files on figshare will be made available approximately one week before the accompanying article so that the supplementary material can be attributed a unique DOI.

Sincerely,
Dr Sarah Brosnan
Editor, Proceedings B

Associate Editor

Comments to Author:

Thank you for your careful revision. Both reviewers are now satisfied with the changes and I am to accept it for publication pending two minor changes as suggested by the reviewers. First, the potential confounds of kinship should be mentioned in the discussion. Second, the use of the term 'strategic' needs to be clarified (or removed).

Reviewer(s)' Comments to Author:

Referee: 2

Comments to the Author(s).

We are happy with the revisions made by the authors with the exception of one minor adjustment that we would recommend needs to be made to increase transparency. As we had asked to include information about genetic relatedness of the dyads, the table in the supplement now reveals that kinship might actually be an issue causing a bias in the sample. While almost all chimpanzee dyads were related, none of the bonobo dyads were. The potential effect of kinship on foodsharing should thus be reviewed in the discussion and mentioned as a potential bias in the cross-species comparison.

Referee: 1

Comments to the Author(s).

This revision satisfactorily addresses most of my issues of the previous version. I also like the additional analysis. There is one remaining point that I recommend to address: While I agree that the proposers (can) act strategically, I do not see this possibility for the responders – at least not in the form as it is used. I outline the problem at two specific text passages.

Abstract, last sentence “Our results suggest that great apes act as rational maximizers in an Ultimatum Game, yet can also act strategically to maximize their payoff depending on their role (proposer vs. responder) and access to alternatives.” First, strategic behavior is not in contradiction or complementing rational behavior. It is rational to take the behavior of other into account. Second, the responder needs no strategic considerations to make. The responder sees what the proposer has done and can respond. Strategic behavior on the side of the responder would only be relevant if the responder takes future interaction into account; for example, caring about a reputation for being tough. Such behavior seems not to be present in the data.

Page 3, “Furthermore, UGs can complement observational studies [15] to help us investigate whether non-human primates use leverage strategically.” It is unclear what “strategic use” of the leverage means. The responders are simply better off if they reject in the case of leverage. My understanding of “strategic use” means that one gives up an opportunity for higher gains in the future. For example, rejection in the UG can be strategic if the game is repeated.

Typo

Page 5: “payofs” in “trials with one of the two payofs being clearly favorable for the responder”

Author's Response to Decision Letter for (RSPB-2021-1937.R0)

See Appendix B.

Decision letter (RSPB-2021-1937.R1)

11-Oct-2021

Dear Dr Sánchez Amaro

I am pleased to inform you that your manuscript entitled "Chimpanzees and bonobos use social leverage in an Ultimatum Game" has been accepted for publication in Proceedings B.

Data Accessibility section

Open Access

You are invited to opt for Open Access, making your freely available to all as soon as it is ready for publication under a CCBY licence. Our article processing charge for Open Access is £1700. Corresponding authors from member institutions (<http://royalsocietypublishing.org/site/librarians/allmembers.xhtml>) receive a 25% discount to these charges. For more information please visit <http://royalsocietypublishing.org/open-access>.

Paper charges

Sincerely,

Appendix A

Dear Dr Sánchez Amaro:

I have now received comments from two reviewers and the Associate Editor, and am writing to inform you that your manuscript RSPB-2021-1480 entitled "Chimpanzees and bonobos use social leverage in an Ultimatum Game" has, in its current form, been rejected for publication in Proceedings B. This action has been taken on the advice of referees, who have recommended that substantial revisions are necessary. However, based on my read of your manuscript as well as the reviewers' and AE's comments, I would be happy to consider a resubmission, provided the comments of the referees are fully addressed. In particular, please pay attention to reviewer 2's comments regarding the analyses and interpretation. With the caveat that I am, as you know, involved in this literature and have my own thoughts on it, I would add to their comments that the discussion of previous work on the UG in chimpanzees could be a bit more nuanced, especially given that your results are in line with the research suggesting that chimpanzees do sometimes change their behavior in UGs and this concordance might contribute to understanding what circumstances lead to this reaction. However, I leave it to you whether and how you wish to address this point. Finally, I agree with reviewer 1 that the manuscript could be simplified which would make it much more accessible to those outside the field. Of course, this is not a provisional acceptance and we will send your revised manuscript back out for review before making a final decision.

Sincerely,

Dr Sarah Brosnan

Editor, Proceedings B
mailto: proceedingsb@royalsociety.org

Authors (A): We would like to thank the Editor the opportunity to resubmit our paper to Proceedings B. We have now addressed all AEs, Reviewer's 1 and 2 comments. In the current version, we discuss in more detail the previous UG literature in relation to our study. In that sense, the introduction has been restructured and shortened to accommodate those changes together with Reviewer 1 comments (Page 3 L 77-101).

In addition, we have slightly changed the framing of the study, now focusing on the most interesting results, i.e., the proposer's decisions. We have also created a new graphical representation of the conditions following Reviewer's 1 advice (Page 5, Figure 1). We have also reformulated our first model to include the proposed interaction between species and condition. The results of the model differ from the previous version, but the main results remain the same (apes distinguish between the control, social with no leverage, and social with leverage conditions).

In our new manuscript, we discuss in much more detail the species differences with a special emphasis on the issues related to our bonobos sample size and their behavior in prosocial trials. (Pages 12 to 14, L 434-512). We include a discussion on bonobos' behavior in light differences in attentional states and in previous findings in inequity aversion studies. However, we would like to highlight that, while it is true that bonobos still maximized in prosocial trials when the responder was present (leverage and no-leverage), we find it inaccurate to claim that they maximized less in prosocial control trials when the difference between the conditions is of 1%. In that sense, Reviewer's 2 calculations were inaccurate, including the plot they provided. Bonobos adjusted their decisions when the partner was present in selfish but not in prosocial trials. We have also re-named prosocial and selfish conditions. Now, we refer to selfish as trials with one of the two payoffs being clearly favorable for the proposer (hence FP) and prosocial as trials with one of the two payoffs being clearly favorable for the responder (hence FR). We have also re-checked our database in light of Reviewer's 2 comments and we found a couple of minor inconsistencies. We have addressed them and we provide the new dataset. Besides the changes in Model 1 due to the introduction of the new two-way interaction, the other main results remain practically the same. Finally, we highlight methodological details missing in the previous version (Page 7 L 250-251, Page 8 L 295-296). We have highlighted all our major changes in the resubmitted document.

Associate Editor
Board Member: 1
Comments to Author:

This manuscript was reviewed by two experts and myself. We all see many positives of the study and the use of novel and comparative cognitive experiments in chimpanzees and bonobos. The goal of understanding how these animals account for the options available to their partner in versions of the ultimatum game is important and has direct relevance to understanding human cognitive evolution. However, there are problems that need to be addressed before the manuscript might be suitable for publication in Proceedings B. Reviewer 1

has several helpful suggestions for reframing and streamlining the manuscript to make the experimental approach and interpretation clearer for the reader. Reviewer 2 notes some inconsistencies with the data and presentation that need to be clarified. In addition, Reviewer 2 notes several areas for clarification and improvement, including providing more details about the experiments and new ways to analyze the data and present the results. Together these reviewer suggestions amount to streamlining the framing but providing more details in the methods and results as well as new ways to analyze and present the data. Reviewer 2 also notes issues with sample size in bonobos which needs to be acknowledged in the Discussion. If these issues can be addressed, I would consider a revised version which will go out for further review, hopefully to the same reviewers.

*A: We would like to thank the Associate Editor for letting us resubmit our work to Proceedings B for another round of reviews. We found both Reviewer's comments very helpful and we have tried to acknowledge all their advices. We have streamlined the paper as suggested by Reviewer 1 and we provided a more nuanced discussion of the previous UG studies as suggested by the Editor. We have also changed Table 1 following Reviewers' 1 suggestion. We have also renamed the selfish and condition variables. We refer to selfish trials as trials with one of the two payoffs being clearly favorable for the proposer (hence FP) and prosocial trials as trials with one of the two payoffs being clearly favorable for the responder (hence FR). We detail the potential differences between bonobos and chimpanzees in light of Reviewer 2 comments and suggestions as well as potential issues related with our sample size. To that end, we have now modified our model 1 to account for the two-way interaction proposed by Reviewer 2, which results in a significant effect of session*condition. However, we find Reviewer 2 comments inaccurate concerning the results of bonobo's choices in control FR/prosocial. They maximized equally between social (with and without leverage) and control trials in the FR/prosocial condition (a difference of 1% between conditions). Nevertheless, it is true that bonobos did not adjust their choices towards the responders in FR/prosocial trials. Bonobos seemed to have an aversion towards selecting 1 (from 1-5 allocation) over 3 (from 3-3 allocation) rewards, even when that could have induced responders to accept their offer. We discuss these results in the discussion section (Pages 12-13, L 446-484). We have highlighted all our major changes in the resubmitted document.*

Reviewer(s)' Comments to Author:

Referee: 1

Comments to the Author(s)

This study presents an experiment chimpanzees and bonobos as subjects. It uses a variant of a mini ultimatum game, in which the responder gets an outside option if she rejects. The outside option is chosen such that if the proposer chooses the kind option, it is in the material self-interest of the responder to accept, and if the proposer chooses the unkind option, it is in the material self-interest of the responder to reject. The results show that both, chimpanzees as well as bonobos, strongly discriminate between the kind and the unkind offer and reject the unkind offer frequently. The bonobos choose the rejection a bit less likely, which could be interpreted that

the bonobo responders are somewhat kinder. The chimpanzee proposers anticipate this behavior and choose the kind offer with a higher probability, while this is not much less the case with the bonobos.

The design is very interesting. The responder behavior shows whether they take the outside option into account. They do it to a high degree. This behavior is not so surprising because the alternatives are directly visible. The proposers have a more difficult task. They have to anticipate the behavior of the responder, and they have to determine the expected benefit of the two alternatives. Because the rejection rates are rather high, it would be expected payoff maximizing (and less risky) to choose the kind option if the outside option is as in the experiment. It is surprising that this is not the case, which means that they take the behavior of the responders only incompletely into account. Further, the treatments vary in whether the kind or the unkind option is fair. Different to human subjects, the apes do not make much of a difference.

A: Thanks for the kind comments. Given our experience with apes and in light of previous social dilemma studies, we do not find surprising that while chimpanzees, and to a lesser extent bonobos, do adjust their offers in relation to responders, they still choose selfishly in a majority of trials. We agree with the Reviewer that bonobos' responder choices seem kinder than chimpanzees. Nonetheless, bonobos proposed more selfish offers than chimpanzees. We have reframed the discussion of our results by highlighting potential issues related with bonobos' sample size as suggested by another reviewer, and new ways to discuss the results (Pages 12 to 14, L 434-512). Furthermore, the other reviewer suggested a modification in model 1 to account for a possible relation between specie and condition (Page 10, L 389-391). This is now significant and we discuss it as well in the new version of the paper.

The experiment is very well set up and contains good control conditions. The statistical analysis and the presentation of the result is clear. However, the motivation does not convince me. The fact that the apes take the "possibility of using social leverage" means only that they take the alternative into account – and this alternative is clearly visible. In my view the study addresses an interesting topic but I disagree with the authors about what is interesting. In my view, the change in the responder behavior is not particularly surprising, also in comparison to the ultimatum game. Nevertheless, the experiment is an interesting setup in order to investigate anticipation of the behavior of others.

A: We understand the reviewer's point and we agree with it. The most interesting result is that proposers take into account the responder's alternative and significantly adjust their decisions (although not perfect as the reviewer previously highlighted). The responders generally choose the maximizing option, with chimpanzees being significantly more likely to choose the alternative when that was the best choice. However, while we agree with the Reviewer in general, we think that responders' choice are still relevant and deserves discussion as well. As the reviewer suggests, apes responses could have motivated proposers, especially chimpanzees, to anticipate and offer lower responses in advance. That is why we have tried to discuss proposer and responder's decisions in relation to one another. In any case, we now emphasize more the proposers' results, for instance in our abstract and at the end of the discussion (Pages 2 L 40-47 and 15 L 563-565).

I recommend refocusing the paper. First, I recommend explaining the experiment in simpler terms. I would present Table 1 earlier and I would (also) present it in a game tree. Further, the terms “selfish condition” and “prosocial condition” are confusing. The conditions are not selfish; only a decision can be selfish. The conditions differ in whether the options favor the proposer or the responder.

A: We agree with the reviewer. We have made an effort to explain the experiment in simpler terms. We have re-done Table 1 according to the reviewer suggestions (Page 5). Finally, we now refer to “selfish” as trials with one of the two payoffs being clearly favorable for the proposer (hence FP) and “prosocial” as trials with one of the two payoffs being clearly favorable for the responder (hence FR) (Page 4 L 137-142). We hope the reviewer agrees with the terminology and all the other changes.

Second, I would develop hypothesis based on the game theoretical predictions. According to this prediction, the responder behavior is not so surprising. In the last paragraph, it is written that “great apes still behaved as rational maximizers but in more nuanced strategic ways than previously stated. For example, responders did not accept any non-zero offer.” I do not see in what respect the behavior is nuanced. The responders usually just took the selfishly better option. But as stated above, the experiment provides an interesting setup to study how the strategic situation is taken into account.

Refocusing the paper could also reduce it somewhat in length.

A: Thanks a lot for the comments. We agree with the main point of the reviewer that apes responder behavior is not so surprising. We have refocused the introduction (hypothesis) and discussion slightly to better represent the fact that responders are mostly acting according to a rational maximizer strategy to solve the UG, in line with all previous responder behaviors in UG with great apes (Page 4 and 5 L 164-167, Page , Page 15 L 563-565).

Referee: 2

Comments to the Author(s)

The authors present an impressive experimental version of the ultimatum game performed by chimpanzees and bonobos to test whether they use social leverage strategically in a feeding context. They show that both species maximize their food reward depending on the presence of leverage. They also find that bonobos overall maximize more than chimpanzees and that there were no differences between the species in patterns of tests or between conditions.

A: We thank the reviewer for the kind words. Below we addressed all the reviewer concerns and comments.

While we agree with the authors that the patterns is likely robust in chimpanzees, we question to some extent the results in the bonobos and therefore the interpretation of the results in light of species differences. Looking at bonobo maximizing behavior across conditions, it looks like they maximize more mainly in the prosocial condition vs chimpanzees, and not so much in the selfish condition.

It also seems that within the prosocial condition, they maximize more in the social condition with or without leverage than they do in the control, which would indicate that they may not understand the set-up very well or that there was potentially some sort of testing bias present. These concerns are explained in more detail below. While the small sample size in bonobos might not allow for three-way interaction effects between species, leverage and conditions, some discussion at the least is needed on this.

A: We mostly agree with the reviewer comments. We agree that our sample size of bonobos is a limitation for statistical analysis. In the new version of the manuscript we discuss in more detail these potential issues (Page 14 L 504-512).

Concerning our models and the treatment of the species and condition variables, we followed the Reviewer suggestions and we included the interaction between condition and specie in model 1. It is now significant and we discuss it in the new version of the manuscript. Most importantly, the main result does not change (that apes tend to offer more when responders are present and have leverage) (Page 10, L 389-391, Pages 12-13, L 446-484) . However, since we did not have any a priori hypothesis suggesting that species would behave differently between conditions, we have not modified our hypothesis in that regard. Overall, bonobos maximized more in prosocial trials than chimpanzees (93% vs 81%). In fact, bonobos and chimpanzees behaved similarly in selfish trials (85% vs 84%). Notice that in the resubmitted version of our manuscript, we refer to selfish trials as trials with one of the two payoffs being clearly favorable for the proposer (hence FP) and prosocial trials as trials with one of the two payoffs being clearly favorable for the responder (hence FR). We are discussing these differences in more detail as the reviewer suggests.

However, we disagree with the claim that bonobos maximized more in the FR/prosocial test conditions with or without leverage compared to FR/prosocial control trials. In the dataset the reviewer had access to before, bonobos maximized in 93.7% of times across conditions. Thanks to the Reviewer comments, we reassessed the dataset completely and we actually found that bonobos maximized slightly less in FR/prosocial control trials, but still in 92.7% of trials. The results did not vary in FR/prosocial social conditions. We conclude that bonobos are maximizing equally in social and control trials within the FR/prosocial condition. A difference of 1% is not enough in our opinion.

Abstract and introduction

Line 41: change “we offer chimpanzees and bonobos responders the possibility” to “we offer chimpanzee and bonobo responders the possibility”

Line 97: but see also:

Oberliessen & Kalenscher, 2019 Social and non-social mechanisms of inequity aversion in non-human animals. *Front. Behav. Neurosci.*

Line 108: I'd recommend rephrasing this sentence. It's not clear from reading this what you mean with “visual cues of estrous over extended periods” give females leverage. I understand that ovulation is hidden in bonobo females because they have a longer period of sexual swelling, and that this is a mechanism that ultimately potentially protects them against male mate guarding but this is not really clear from this sentence. I'm also not sure this is a great example of social leverage tactics since it's not really something the females actively decide to do but rather a mechanism that developed through years of natural/sexual selection.

A: Thanks for all the comments. We have made substantial changes in the introduction, following the suggestions of the other reviewer and the Editor. The section on leverage in wild populations has been removed. The idea is briefly mentioned in (Page 2, L 73-74).

Line 113 & 141: In the light of the ability to “consciously” use the leverage: Did you observe any begging, harassment behavior or other communication from the responder before the choice by the proposer that may have influenced the proposer’s choice? To what extent did they pay attention to each other or to the choices they made? Did you observe any reaction after the responder received the offer?

A: That is a very interesting topic. The first author, who tested the apes, does not recall any observable signs of begging or harassment. The food was baited when they were both visually attentive so they were, as far as we can tell, aware of each others’ choices. We did not observe any agonistic reaction towards the responder. However, from time to time they reacted aggressively towards the experimenter when he had to retrieve the food from the apparatus (e.g. when the proposer’s offer was not accepted). For future studies, the first author has designed an apparatus to finalize trials and clear rejected constellations without the need of human presence.

Line 174: The view that bonobos are more tolerant than chimpanzees is a bit outdated and dependent on what is considered as a measure for tolerance. It is true that they have higher levels of social tolerance in the form of intercommunity interactions (reduced rates of aggression and lethal aggression) but when it comes to feeding tolerance, there is a lot of inconsistency in the outcomes of different experiments. The studies cited are indeed examples of tolerant food sharing but there is plenty of additional literature claiming the reverse or at least equal feeding tolerance and large within-species variation that exceeds between-species variation. This is also relevant for your discussion where you claim that bonobos are less competitive around food and more willing to share. For example see:

Amici, F., Call, J. & Aureli, F. Aversion to violation of expectations of food distribution: The role of social tolerance and relative dominance in seven primate species. *Behaviour* 149, 345–368 (2012)

Bullinger, A. F., Burkart, J. M., Melis, A. P. & Tomasello, M. Bonobos, *Pan paniscus*, chimpanzees, *Pan troglodytes*, and marmosets, *Callithrix jacchus*, prefer to feed alone. *Anim. Behav.* 85, 51–60 (2013)

Cronin, K. A., De Groot, E. & Stevens, J. M. G. Bonobos show limited social tolerance in a group setting: A comparison with chimpanzees and a test of the relational model. *Folia Primatol.* 86, 164–177 (2015).

Jaeggi, A. V., De Groot, E., Stevens, J. M. G. & Van Schaik, C. P. Mechanisms of reciprocity in primates: Testing for short-term contingency of grooming and food sharing in bonobos and chimpanzees. *Evol. Hum. Behav.* 34, 69–77 (2013).

Jaeggi, A. V., Stevens, J. M. G. & Van Schaik, C. P. Tolerant food sharing and reciprocity is precluded by despotism among bonobos but not chimpanzees. *Am. J. Phys. Anthropol.* 143, 41–51 (2010).

A: The reviewer is right. Although we would prefer to not change the core aspects of our initial hypothesis (that if any, chimpanzees might behave more strategically than

bonobos), we are now considering the alternative view (unclear species differences in relation to feeding tolerance) highlighted by the reviewer, and in fact we comment on this in our hypothesis and beginning of the discussion (Page 4, L 157-160 and Page 12, L 434-445).

Methods

Line 187-192: I'd add some information about relatedness for the remaining dyads, I don't think it's an issue here but it's good to add, especially given the small sample size and the important role kinship could play on food sharing.

A: Thanks, we now report on their relatedness (Table ESM).

Line 249-260: To me the figures are a bit confusing, likely because of the complex set-up. Is there a way to make the rewards stand out more? I realize they are already a bright green but it's still a bit hard to tell.

Line 267-270: Can you add here why you did this test? What came out of the preference tests? When was decided whether subjects were able to discriminate between quantities? Did they need to reach a specific criterium?

L282 Double space between "... apparatus. Hence..."

L297 What was the criterium that subjects had to pass before proceeding to the next test phase?

L317: "Ten apes... successfully advanced to the test phase" but what does this mean? The validity of the outcome of this study depends on this phase of the pre-test training. Therefore, a more detailed description of this part of the study is necessary.

A: Thanks for all these comments. The reviewer is absolutely right. We did not clarify the criteria to advance from one test to another in the previous manuscript. In the preference test we wanted to be sure that they could distinguish between different food quantities (later used in the test) and that they would prefer to maximize rewards. They had to choose the highest amount in at least 7 out of 8 trials for two consecutive sessions. The criteria was essentially the same in the next two pre-test phases. In pre-test one they had to maximize rewards by choosing 4 over 2 rewards, and in pre-test two they had to choose 2 over 1 or 3 over 2 rewards depending on the condition presented in at least 7 out of 8 trials for two consecutive sessions. We have included all these information in the new main manuscript (Page 7 L 246-251, Page 8 L 295-296).

Line 390: typo, change "." to "," after session. Also, you tested for species effects but there is no mention of this in your statistics section in the main manuscript. Did you test for species by condition/test interaction effects? This would be key to interpret your results.

A: Sorry for the confusion. We reached the word limitation in Proceedings B and, unfortunately, we transferred too much to the ESM. We, indeed, tested for species differences in interaction with leverage (leverage yes, no and control) in our model 1. In addition, for our model 1 we have now tested species in interaction with condition as suggested by the reviewer. We clarify these statistical details in the new main manuscript (Page 10, L 369-373).

Results and discussion

Can you show in the figures which groups differed significantly from others? Either below in the text or in the figure itself using asterisks. I think figure 2 is a bit distracting since it does not show the results of your analysis: neither the species effect, which lumps the data from all leverage levels together, or leverage effect, which lumps data from both species together.

A: We are now highlighting in text the significant differences in the figure captions. Our first plot has changed completely to document the main effect of leverage. We agree that the previous combination of species and leverage main effect (not interaction) was confusing.

Line 457: It's unclear why your results are in line with higher feeding tolerance in bonobos. Isn't the fact that they always maximize more in line with lower feeding tolerance? You do mention later that they do not necessarily behave more prosocially but maybe the fact that they do not distinguish between the test conditions means they did not really understand?

A: This is an interesting point for discussion. From the perspective of the proposer, the reviewer might be right. Bonobos maximize more than chimpanzees by offering less to the responder, in line with the despotism claim during food sharing put forward by Jaeggi et al. 2010. But the responder actually accepts these offers so that both can feed. Furthermore, as stated in the study, while it is true that bonobos did not distinguish between test conditions (possibly enhanced by the fact that responders accepted those offers more often than in chimpanzees), they distinguish between control and test trials in FP/selfish condition and overall maximized more than chimpanzees in control trials. In any case, the reviewer is right and bonobos might have not grasped the strategic aspect of the task compared to chimpanzees. One possibility is that bonobos focused mainly on the food present for the proposer. Another one is that they were more inequity averse than chimpanzees. We discuss these possibilities in length in the new discussion (Pages 12 to 14, L 434-512).

Note though that bonobos did not show specific biases during the pre-test phases and they had to reach the same performance criteria established for chimpanzees. The only exception was Luiza, a bonobo that was particularly motivated to accept offers and reject alternatives across FP/selfish and FR/prosocial conditions. In any case, that was not due to a side bias since she still accessed the alternative in almost half of the test trials with leverage and she did not show any bias between the two apparatus sides when acting as a proposer.

Did you consider analyzing a species by condition interaction effect? When looking at the proportion of maximized choices across conditions and species, the higher levels in bonobos compared to chimpanzees seem to be present primarily in the prosocial condition in test trials, but not control trials: bonobos seem to maximize less in the prosocial condition in the absence of a responder. This is surprising if bonobos are expected to take others behavior into account. This contrasts with line 398-404.

A: We followed the Reviewer's advice and we include the interaction, which showed a significant effect in the direction suggested by the Reviewer. We agree that the main difference between bonobos and chimpanzees lies in FR/prosocial condition in test trials. However, in FR/prosocial control trials bonobos maximized 7% more than chimpanzees. The same cannot be said when bonobo FR/prosocial control trials are

compared with bonobo FR/prosocial test trials. Bonobos practically act the same way: 92.7% of maximizing choices in FR/prosocial control trials and 93.7% in FR/prosocial test trials.

Moreover, if bonobos prefer 3 over 1 grape, why do they show such low maximizing levels in the prosocial trials in the control sessions, not showing the self-maximizing offers (Line 468)?

A: We are not sure we understand the Reviewer's point here. Bonobos maximized in almost 93% of the FR/prosocial trials in the control sessions. In our opinion, they clearly showed self-maximization.

Why would a subject choose the '1-5' option in the prosocial control condition at all?

A: Bonobos were confronted with 96 of those control trials and they reliably preferred 3 over 1 in 89. It is unclear (possibly distraction) why they chose the 1-5 option in the remaining 7 trials.

(The same is true for the '3-3' option in the selfish trials in the control condition, which is even lower for bonobos). The figure that we attached to this document is based on the data provided by the authors with the manuscript and reflects this issue more clearly.

A: Thank you the comments. However, we have noticed two inconsistent values. In the no leverage bonobos FR/prosocial trials, they did not maximize 100% of times, but 93.75%. In control bonobos FP/selfish trials they did not maximize in 85% but in 93.75% of trials. In general, bonobos maximized in over 93% of control trials across conditions (FR and FP), very similar values to what they scored in FR/prosocial test trials with and without leverage. Here we provide a new figure based in the attached dataset:

Line 477 “Bonobos accepted more unequal offers” and line 498 “maximized more regardless of the responders options”. In combination with the previous comment, this raises the question whether bonobos did understand task contingencies.

A: We agree with the reviewer, and we address this possibility in more detail: In contrast to chimpanzees, bonobos may have not grasped the impact of the alternative (no clear differences between trials with and without leverage in bonobos) and they seem to show an aversion towards one (from 1-5) over three (from 3-3) rewards preventing them to adjust their offers in FP/prosocial trials when the responder was present. (Page 12 to 14 L 446-484, L 500-512). However, we would like to emphasize again that bonobos did not make significant mistakes in control trials as shown in our dataset and attached figure. Their performance was higher than chimpanzees. Nevertheless, we also acknowledge our limited sample size when interpreting the results (Page 14, L 504-512).

Even though no bias towards a specific constellation across conditions was found, other aspects could have caused the relatively small differences in outcome (like task complexity, local enhancement due to presence of the conspecific on the right side). Did you test whether any side-biases were present in the actual test phase of the experiment? e.g did subjects that had to be retested in the first pre-test phase show higher or lower maximizing scores or biases? Or did (some of) the individuals show a slight left-sliding preference (cfr lower choices for alternative offer)?

A: Thanks for the comments. We assessed in detail for the presence of side biases. We did not find any side bias towards one of the sides of the study. The highest side bias towards the left choice in the proposer side was of 58% in a chimpanzee. Subjects that had to be retested in the two pre-test phases did not show particular side biases during the test. We discuss these points in the new discussion (Page 13, L 468-474).

What did the set-up look like on the left-side in the no-leverage condition? Although there was no food baited on the alternative food compartment, did the responders still have to slide the door to the left to access the offer of the proposer? Could there be any carry-over effect of these no-leverage sessions on the leverage-sessions that may have resulted in lower overall alternative choices by bonobos? Or did bonobos prefer to keep their distance from the conspecific resulting in a preference for the right sight of the apparatus, i.e. the side of the offer by the proposer?

A: Thanks for the comments. When there was no leverage, apes could still access the empty alternative. Only one chimpanzee accessed it once by mistake. When no leverage was presented, they overwhelmingly accepted the offer. We counterbalanced sessions with and without leverage. We assessed, for instance, whether Luiza developed a tendency across sessions to choose the alternative given her low preference for that option, but that was not the case. Apes were at the same distance from the responder regardless of their choice.

Line 486-488: This is in contrast to what was previously said about bonobos being risk averse and chimpanzees choosing the riskier option, in line with differences in their feeding ecology. Maybe this deserves some discussion here as well:

Heilbronner, S.R., Rosati, A.G., Stevens, J.R., Hare, B., Hauser, M.D. (2008). A fruit in the hand

or two in the bush? Divergent risk preferences in chimpanzees and bonobos. *Biology Letters*, -1(-1), -1-1. DOI: 10.1098/rsbl.2008.0081

A: Thanks. This is an interesting comment but due to the general changes in the discussion, we no longer discuss this point. However, one could also imagine that chimpanzees betting on the less self-maximizing choice is a risky strategy as well, because they still depend on the partners' acceptance. Perhaps bonobos' strategy might have been less risky given their lower propensity to access alternatives.

Line 523-524: How does the amount of maximization differ between the control trials in sessions 1-2 and 11-12? The data do suggest an increase in maximization, hence possible task comprehension?

A: It increases around 10% from 85 to 95%. That is the main reason why we preventively ran our model 1 without control trials. Without control trials, no session effect was found.

Supplement

Figure S2: Squares should represent bonobo-dyads, which is not true for Yasa-Kuno: represented by a pink circle in stead of a pink square; Sandra-Frodo, a chimpanzee dyad, should be represented by a circle.

A: Thanks a lot. We have re-done the figure.

Data

check data lines

- 129: selfish condition control– choice subject 1: corresponds with prosocial condition?
- 864: prosocial condition – choice subject 5, choice partner 2 corresponds with selfish condition
- 392; 739; 1125: Alternative column is empty

How did you take trials into account where subjects and partners made a choice but subjects did not eat the food. It seems that especially for the dyad Kuno-Yasa this happened 8 times. Did you consider these trials in your analyses? What would these choices mean e.g choosing the maximized option in the selfish condition with a partner that agreed with the offer but then the subject decided to not consume the food items? What happens if you exclude them from the analyses?

Please check whether these suggestions concerning the data influence analyses and results.

A: Thanks for the comments. We found a couple of inconsistencies in the data. We are submitting a new datafile. The new results does not change the previous interpretations. Concerning the last point, this was a typo. The bonobo Kuno always ate the food when Yasa accepted the offer. The result does not influence the model since we never used food consumption in any of our models.

Appendix B

Dear Dr Sánchez Amaro

I am very pleased to inform you that your manuscript RSPB-2021-1937 entitled "Chimpanzees and bonobos use social leverage in an Ultimatum Game" has been accepted for publication in Proceedings B. Personally, I really like your study and think that your paper will be a strong contribution to the literature. Please note that the reviewers have recommended a few minor revisions to improve the clarity of your manuscript (see their comments, appended below). In addition, I wonder if in Figure 1 it might clarify to indicate that the "Responder Choice" is really their ideal choice to maximize their outcome, not what they picked in the study (and I agree with the reviewers that the inclusion of the figure makes it much easier to follow the method for those who won't be as familiar with the general paradigm, so thank you for doing so). I invite you to respond to the referee(s)' comments and revise your manuscript. Because the schedule for publication is very tight, it is a condition of publication that you submit the revised version of your manuscript within 7 days. If you do not think you will be able to meet this date please let us know.

Authors: We are very happy to get our paper accepted in Proceedings B. We are addressing the two comments by the reviewers in our answer below. We have also addressed the Editor comment. We now frame responder choices as "responder ideal choices" and we describe what that means in the figure caption (L 169-171).

Associate Editor

Comments to Author:

Thank you for your careful revision. Both reviewers are now satisfied with the changes and I am to accept it for publication pending two minor changes as suggested by the reviewers. First, the potential confounds of kinship should be mentioned in the discussion. Second, the use of the term 'strategic' needs to be clarified (or removed).

Authors: We agree with the reviewers and we are addressing their comments in the response below.

Reviewer(s)' Comments to Author:

Referee: 2

Comments to the Author(s).

We are happy with the revisions made by the authors with the exception of one minor adjustment that we would recommend needs to be made to increase transparency. As we had asked to include information about genetic relatedness of the dyads, the table in the supplement now reveals that kinship might actually be an issue causing a bias in the sample. While almost all chimpanzee dyads were related, none of the bonobo dyads were. The potential effect of kinship on foodsharing should thus be reviewed in the discussion and mentioned as a potential bias in the cross-species comparison.

Authors: The reviewer is right. Given the small sample size we could not really control for kinship but we see the potential issue and we briefly mention it in the discussion (L 499-503).

Referee: 1

Comments to the Author(s).

This revision satisfactorily addresses most of my issues of the previous version. I also like the additional analysis. There is one remaining point that I recommend to address: While I agree that the proposers (can) act strategically, I do not see this possibility for the responders – at least not in the form as it is used. I outline the problem at two specific text passages.

Abstract, last sentence “Our results suggest that great apes act as rational maximizers in an Ultimatum Game, yet can also act strategically to maximize their payoff depending on their role (proposer vs. responder) and access to alternatives.” First, strategic behavior is not in contradiction or complementing rational behavior. It is rational to take the behavior of other into account. Second, the responder needs no strategic considerations to make. The responder sees what the proposer has done and can respond. Strategic behavior on the side of the responder would only be relevant if the responder takes future interaction into account; for example, caring about a reputation for being tough. Such behavior seems not to be present in the data.

Page 3, “Furthermore, UGs can complement observational studies [15] to help us investigate 74 whether non-human primates use leverage strategically.” It is unclear what “strategic use” of the leverage means. The responders are simply better off if they reject in the case of leverage. My understanding of “strategic use” means that one gives up an opportunity for higher gains in the future. For example, rejection in the UG can be strategic if the game is repeated.

Typo

Page 5: “payofs” in “trials with one of the two payofs being clearly favorable for the responder”

Authors: We agree with the reviewer and we have changed both paragraphs accordingly. We think that responders used their leverage to maximize their own gains, and that in turn influenced proposers future behavior since the game was repeated (L 44-47; 73-74).